# Understanding the Soupability of Documents in State Space Models

## Abstract

We investigate whether hidden states from Structured State Space Models (SSMs) can be merged post hoc to support downstream reasoning. Inspired by model souping, we study document souping, a strategy where documents are encoded independently, and their representations are pooled, via simple operations like averaging, into a single context state. This approach enables modular encoding and reuse without reprocessing the full input for each query. We demonstrate that fine-tuned Mamba2 models with souped representations achieve competitive or superior performance across multi-hop QA, sparse retrieval, and long-document reasoning tasks compared to the standard monolithic encoding approach. For example, on the RACE and QuALITY benchmarks for long document question answering, this method substantially outperforms a traditional concatenation approach. Crucially, this modular design scales to hundreds of documents—we test up to 256—while delivering substantial savings in inference cost, unlocking new possibilities for large-scale corpus reasoning.

## 1 Introduction

Many real-world NLP tasks, such as multi-document question answering, scientific summarization, and legal analysis, require reasoning over entire corpora rather than individual long documents. These tasks demand flexible integration of information distributed across sources, as well as the ability to dynamically update, prune, or recombine input subsets. Yet today's language models remain poorly suited for this kind of modular document reasoning.

Transformer-based models (Vaswani et al., 2017), despite their success, face prohibitive $\mathcal{O}(L^2)$ attention costs that make full corpus encoding expensive and inflexible. Structured State Space Models (SSMs) offer a promising alternative: architectures like Mamba (Gu & Dao, 2024) and Mamba2 (Dao & Gu, 2024) process sequences in linear time, compressing them into fixed-length hidden states. The linear recurrence central to these models provides the core intuition for our work: we hypothesize that linear operations on these states, such as averaging, will produce meaningful composite representations. This architectural feature is key, as we later show empirically that it makes SSMs uniquely suited for the state-souping setup we study, an approach that proves ineffective in standard Transformers. However, even these efficient architectures are typically deployed using a monolithic encoding strategy: concatenating all documents into a single sequence before processing. This approach inherits a critical weakness: any modification to the corpus, even changing a single document, requires re-encoding the entire input from scratch and prevents the reuse of document representations. This re-encoding requirement becomes prohibitive at scale—processing hundreds or thousands of documents repeatedly for each query.

This brittleness suggests a fundamental mismatch: we want modular, reusable representations, yet current approaches still rely on monolithic, use-once encodings. In this work, we explore an alternative inspired by model souping (Wortsman et al., 2022), which merges finetuned checkpoints by averaging their parameters. Recent work has begun to push this idea to hidden states: Pióro et al. (2024) show that in-context *skills* can be stored as recurrent states, retrieved, and linearly mixed, while Liu et al. (2025) derive a theoretically well-motivated operator for composing SSM states that approximates a permutation-invariant version of full concatenation and include a simple *souped*-state baseline. However, these approaches leave open a key question: can *factual knowledge* spread

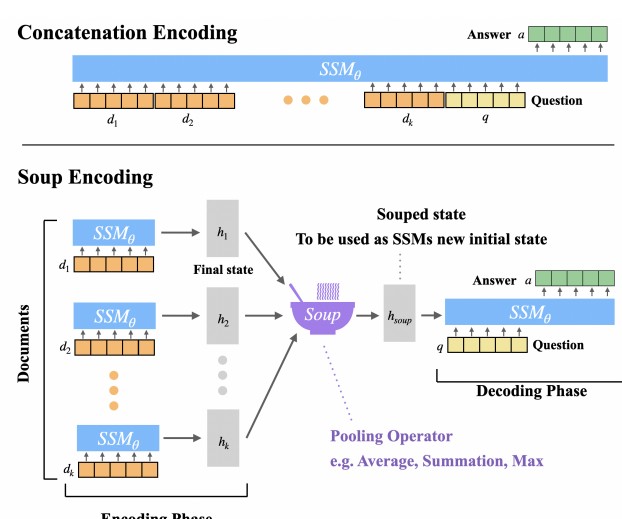

Figure 1: **Computation Graphs for Corpus Encoding.** *Top:* In traditional concatenation-based encoding, all documents $\{d_1, \ldots, d_k\}$, the query $q$, and answer $a$ are flattened into a single input sequence and processed end-to-end by an SSM. This requires joint re-encoding for every change to the input. *Bottom:* In the state-souping approach we study, each document $d_i$ is encoded independently by a shared SSM, producing per document hidden states $\{h_1, \ldots, h_k\}$ which are pooled into a single representation $h_{\text{soup}}$ (e.g., via sum or average). This pooled state is then used, alongside the query $q$, to drive downstream prediction. The design supports parallel encoding, modular reuse, and post hoc corpus composition.

across long documents and corpora be reliably souped and composed to support real multi-document reasoning, and what failure modes arise when one attempts to do so?

We address this question from a complementary, empirically driven perspective. If SSMs encode individual documents into fixed-length hidden states, can those representations be *merged post hoc* while still supporting downstream tasks such as multi-hop QA and long-document reading comprehension? We refer to this property as **soupability**: the ability of independently encoded document representations to be pooled while preserving the information needed for multi-document reasoning. Our aim is to test souping capabilities for realistic multi-document and long-context settings, analyze when it fails, and what training and pooling regimes make it effective in practice.

To study soupability concretely, we operationalize it as **corpus encoding via state souping**, illustrated in Figure 1. Each document is independently encoded by a shared SSM to produce per-document hidden states, which are then pooled using simple commutative operators (e.g., averaging) into a single representation that conditions the decoder alongside the query. This construction exposes both the strengths and weaknesses of souping: naive pooling generally does *not* work out of the box, but with appropriate finetuning and pooling choices we find that factual content in larger documents and corpora can be reliably souped and composed, yielding performance competitive with joint encoding on challenging multi-hop benchmarks. At the same time, the approach retains a key advantage of modularity: documents can be encoded once, cached, and later combined in arbitrary subsets, enabling efficient corpus updates and dynamic retrieval without re-encoding.

We use Mamba2 as our representative SSM because its core ingredients are shared across modern SSM families, including S4 (Gu et al., 2022) and S5 (Smith et al., 2023) variants. Moreover, it is the only large-scale SSM without a hybrid architecture. Our approach depends on the linear recurrent scan that produces encoder hidden states, not on details unique to Mamba2. The properties that enable souping, namely independent encoding, linear aggregation, and conditioning on a pooled state, should therefore carry over to other SSM architectures with comparable state representations.

We find that Mamba2 encodes representations with strong soupability. With full encoder-decoder finetuning, our souping-based architecture supports multi-hop QA, sparse retrieval, and long document understanding, often matching or surpassing traditional joint encoding approaches. This design unlocks a new inference workflow where corpora can be encoded once, cached, and reassembled dynamically, enabling scalable, retrieval-based reasoning without repeated full sequence processing.

**Contributions.** We systematically investigate the soupability of document representations in SSMs using Mamba2-2.7B (Dao & Gu, 2024) and Mamba2-8B (Waleffe et al., 2024). Our contributions are: (1) We provide the first systematic empirical study of **document souping**, its behavior, and

limitations. (2) We demonstrate that finetuned Mamba2 models with souped representations can support multi-hop reasoning on HotpotQA (Yang et al., 2018), with performance comparable to joint encoding. (3) We analyze soupability across diverse tasks, including long document QA (RACE (Lai et al., 2017) and QuALITY (Pang et al., 2022)) and sparse retrieval. (4) We compare pooling strategies and show that soupability is robust to operator choice. (5) We show that this modular souping approach is uniquely suited to SSMs, as an analogous technique fails in standard Transformer architectures.

## 2 METHODS

We investigate when document representations from Structured State Space Models (SSMs) can be *merged post hoc*, using simple commutative operations like averaging or summation, while still preserving the information needed for downstream tasks. We refer to this property as **soupability**.

### 2.1 BACKGROUND AND NOTATION

Structured State-Space Models (SSMs) are a class of sequence models that compress long inputs into fixed-size hidden representations through linear recurrence mechanisms. Unlike attention-based models, SSMs offer subquadratic compute and memory scaling, making them well-suited for long-context settings.

We denote the encoder component of an SSM as a function $\text{SSM}_\theta$, which maps a document $d$ to a sequence of layer-wise hidden states $\{h^{(1)}, \ldots, h^{(L)}\}$. Each $h^{(l)} \in \mathbb{R}^d$ summarizes the input up to layer $l$, and $L$ denotes the total number of layers. This encoder is applied to the entire input sequence jointly, and the resulting states are passed to a decoder for generation or prediction.

In this work, we leverage the layerwise structure of SSMs to explore whether documents can be encoded independently and later recombined in a modular fashion. Our central question is whether these hidden states, computed separately per document, can be *pooled* into a single representation suitable for downstream reasoning. Given their foundation in linear recurrence, we hypothesize that a linear combination of these states could form a meaningful composite representation. We refer to this approach as **corpus encoding via state souping**.

### 2.2 CORPUS ENCODING VIA STATE SOUPING

The corpus encoding strategy we study is illustrated in Figure 1 and formalized in Appendix A.3. Given a set of documents $\{d_1, \ldots, d_k\}$ and a query $q$, each document is passed independently through a shared SSM encoder, yielding a set of hidden states $\{h_1^{(l)}, \ldots, h_k^{(l)}\}$ at each layer $l$. These are then combined using a commutative pooling operator, typically elementwise average, sum, or max, resulting in a single pooled state $h_{\text{soup}}^{(l)}$ for each layer.

We optionally explore unit normalization, applied either before pooling (to each $h_i^{(l)}$) or after pooling (to $h_{\text{soup}}^{(l)}$):

$$\tilde{h}_i^{(l)} = \frac{h_i^{(l)}}{\|h_i^{(l)}\|}, \quad h_{\text{soup}}^{(l)} = \text{normalize}\left(\sum_{i=1}^{k} \tilde{h}_i^{(l)}\right).$$

Once pooled, these hidden states $\{h_{\text{soup}}^{(1)}, \ldots, h_{\text{soup}}^{(L)}\}$ are injected into the decoder alongside the query $q$. The decoder produces the answer $\hat{y}$ conditioned on the souped representation:

$$\hat{y} = \text{SSM}_\theta\left(q \mid \{h_{\text{soup}}^{(l)}\}_{l=1}^{L}\right).$$

This design enables efficient parallel encoding, modular document reuse, and flexible corpus reconfiguration at inference time. Because the encoder processes each document in isolation, representations can be cached and reused across multiple queries, dramatically reducing redundant computation.

To support training of this architecture, gradients must propagate through multiple independent document encoders. Without memory optimizations, this would require storing intermediate activations for all $k$ documents, leading to memory growth linear in $k$. To address this, we apply **activation**

| Method | Test on $2$ gold + $(n-2)$ distractors | | |
|---|---|---|---|
| | 2 | 5 | 10 |
| **Pretrained 8B (No Finetuned)** | | | |
| Concat | 15.4 / 26.3 | 8.5 / 20.2 | 5.0 / 15.7 |
| Soup w/ Average | 8.7 / 12.7 | 2.6 / 5.0 | 1.7 / 3.6 |
| **Decoder-Only Finetuned 8B** | | | |
| Soup w/ Average | 51.8 / 66.4 | 38.8 / 51.7 | 28.0 / 39.4 |
| **Encoder-Decoder Finetuned 8B** | | | |
| **Full Finetuned - Average Pooling With & Without Norms** | | | |
| Soup w/ Average | 55.8 / 69.8 | 47.8 / 61.3 | 38.7 / 50.9 |
| Average + Norm Before | 35.9 / 47.8 | 47.8 / 60.9 | 38.1 / 50.2 |
| Average + Norm After | 50.4 / 65.2 | 42.3 / 55.6 | 33.4 / 44.9 |
| Average + Norm Before & After | 6.9 / 10.7 | 42.2 / 54.7 | 33.8 / 45.6 |
| **Full Finetuned - Summation Pooling With & Without Norms** | | | |
| Soup w/ Sum | 55.1 / 69.4 | 44.2 / 57.4 | 25.0 / 36.0 |
| Sum + Norm Before | 8.8 / 13.6 | 8.1 / 13.0 | 4.6 / 9.4 |
| Sum + Norm After | 51.9 / 66.1 | 43.6 / 56.6 | 34.7 / 46.2 |
| Sum + Norm Before & After | 10.2 / 16.3 | 40.3 / 52.9 | 33.2 / 44.8 |
| **Full Finetuned - Max Pooling Without Norms** | | | |
| Soup w/ Max | 52.3 / 65.6 | 39.1 / 51.6 | 28.9 / 40.5 |

Table 1: HotpotQA performance (Exact Match / F1) for Mamba2-8B models trained on 5 documents (2 gold + 3 distractors) and evaluated on $n$ documents, each with 2 gold and $(n-2)$ distractors. We compare pretrained models, decoder-only finetuning, and encoder-decoder finetuning across different pooling strategies (average, sum, max), with optional normalization applied before and/or after aggregation. Underlined entries indicate evaluations where the number of test-time documents matches the training configuration. We observe that decoder-only finetuning improves performance by learning to interpret fixed souped states, and full encoder-decoder finetuning yields the best results by also learning to produce mergeable representations. Across all configurations, simple averaging without normalization emerges as the most stable and effective aggregation method.

**checkpointing** at the document level: forward activations are recomputed during the backward pass, enabling constant memory usage regardless of corpus size. This allows us to scale finetuning to wide or deep document sets efficiently.

## 2.3 Evaluation Dimensions

To characterize when corpus encoding via state souping is effective, we organize our analysis around two dimensions: model capacity and corpus structure.

**Model capacity** concerns the architectural and training properties that affect soupability. We ask: Are pretrained SSMs inherently soupable, or must they be finetuned to interpret pooled states? Does soupability improve with model size or hidden state dimensionality? And how well do models generalize across soup sizes, for example, when asked to merge more documents at test time than during training?

**Corpus structure** examines how input organization affects pooling success. We study whether long, contiguous documents can be segmented and recomposed via souping, or whether this method is better suited to independently authored texts. We also test whether souped representations preserve the dependencies needed for multi-hop reasoning, where answering a query requires synthesizing information from multiple documents.

Together, these questions guide our exploration of soupability as a flexible and scalable alternative to serial encoding for corpus-level reasoning in state space models.

## 3 Experiments

We conduct a comprehensive set of experiments to evaluate document souping across a range of tasks and settings. This section outlines our experimental design, detailing the datasets, baselines, and implementation specifics used in our analysis.

### 3.1 Tasks and Datasets

We evaluate state souping across a range of long-context reasoning tasks that test different dimensions of soupability: single-hop vs. multi-hop inference, monolithic vs. multi-document structure, and sparse signal detection in distractor-heavy inputs.

Table 2: HotpotQA results for Mamba2-8B evaluated on 2 gold documents plus $(n-2)$ distractors. We compare concat-based and soup-based training across a range of finetuning configurations: QA-only (no documents), 2-gold only, and 2-gold + distractors. Each model is evaluated at multiple soup sizes, and underlined entries mark evaluations that match the training number of documents.

| Method | Test on 2 gold + $(n-2)$ distractors | | | | | | | | | |
|---|---|---|---|---|---|---|---|---|---|---|
| | $n = 0^{\dagger}$ | 2 | 3 | 4 | 5 | 6 | 7 | 8 | 9 | 10 |
| **Pretrained, no finetune** | | | | | | | | | | |
| Concat | 2.4 / 4.7 | 15.4 / 26.3 | 11.1 / 22.5 | 9.7 / 21.6 | 8.5 / 20.2 | 7.2 / 18.9 | 6.4 / 17.1 | 5.5 / 16.5 | 6.0 / 17.1 | 5.0 / 15.7 |
| Soup w/ Average | – | 8.7 / 12.7 | 5.3 / 8.3 | 3.7 / 6.4 | 2.6 / 5.0 | 2.2 / 4.5 | 2.3 / 4.5 | 2.0 / 4.0 | 1.7 / 3.9 | 1.7 / 3.6 |
| **Full Finetuned on $n = 0$ $gold + 0$ $distractors$ (QA-only)** | | | | | | | | | | |
| Concat | 18.8 / 27.4 | 52.2 / 67.4 | 48.0 / 62.3 | 44.9 / 58.5 | 42.0 / 55.1 | 38.5 / 51.5 | 36.6 / 49.5 | 36.3 / 48.8 | 34.8 / 46.9 | 33.0 / 45.0 |
| **Full Finetuned on $n = 2$ $gold + 0$ $distractors$ documents** | | | | | | | | | | |
| Concat | – | 56.0 / 70.3 | 51.3 / 65.1 | 46.2 / 59.8 | 43.6 / 57.3 | 40.1 / 53.5 | 37.6 / 50.5 | 36.2 / 48.1 | 34.4 / 46.4 | 34.4 / 45.8 |
| Soup w/ Average | – | 57.1 / 70.9 | 51.1 / 64.4 | 46.7 / 59.7 | 43.0 / 56.0 | 39.2 / 52.1 | 36.2 / 48.5 | 34.2 / 46.0 | 32.2 / 44.0 | 30.6 / 42.2 |
| **Full Finetuned on $n = 2$ $gold + 3$ $distractors$ documents** | | | | | | | | | | |
| Concat | – | 57.1 / 71.3 | 54.4 / 68.3 | 51.9 / 66.3 | 49.0 / 63.1 | 47.9 / 61.5 | 45.5 / 59.0 | 44.4 / 57.8 | 42.9 / 55.4 | 41.4 / 54.1 |
| Soup w/ Average | – | 55.8 / 69.8 | 53.0 / 66.5 | 50.0 / 63.6 | 47.8 / 61.3 | 45.3 / 58.7 | 43.9 / 56.6 | 40.9 / 53.6 | 39.5 / 52.2 | 38.7 / 50.9 |
| **Full Finetuned on $n = 2$ $gold + 5$ $distractors$ documents** | | | | | | | | | | |
| Concat | – | 54.6 / 69.4 | 52.6 / 67.1 | 50.8 / 64.4 | 49.1 / 63.1 | 47.8 / 61.4 | 45.1 / 58.7 | 44.3 / 57.3 | 42.4 / 55.8 | 41.6 / 54.3 |
| Soup w/ Average | – | 55.5 / 69.4 | 52.2 / 66.0 | 50.3 / 63.9 | 48.1 / 61.5 | 46.2 / 59.5 | 45.0 / 57.6 | 43.2 / 55.8 | 41.8 / 54.2 | 40.3 / 52.8 |
| **Full Finetuned on $n = 2$ $gold + 8$ $distractors$ documents** | | | | | | | | | | |
| Concat | – | 55.0 / 69.5 | 52.1 / 66.7 | 48.9 / 63.5 | 48.0 / 62.5 | 46.9 / 60.8 | 45.2 / 59.0 | 43.2 / 57.6 | 42.0 / 56.0 | 42.5 / 56.0 |
| Soup w/ Average | – | 54.8 / 68.7 | 52.6 / 65.9 | 50.1 / 63.5 | 47.7 / 60.8 | 45.8 / 58.5 | 44.7 / 57.0 | 43.5 / 55.8 | 41.6 / 53.7 | 40.8 / 52.9 |

$^{\dagger}$ $n = 0$ corresponds to a no-context setting where the model receives only the query (i.e., 0 gold documents and 0 distractors).

**Multi-Doc QA**   We study multi-document question answering in two distinct regimes. For single-hop QA, we use the QA subset of the RULER dataset (Hsieh et al., 2024), which augments SQuAD-style (Rajpurkar et al., 2016) questions for long-context evaluation. Each question is answerable from a single gold document placed within a large set of distractors, allowing us to isolate how well the model can identify and preserve localized information in souped representations. For multi-hop QA, we turn to HotpotQA (Yang et al., 2018), a benchmark requiring compositional reasoning across multiple Wikipedia paragraphs. Each question demands integration of evidence from at least two documents, providing a direct test of whether souped hidden states preserve the relational structure needed for multi-hop reasoning.

**Long Doc QA**   We also evaluate on long-document QA tasks, where inputs are single extended narratives rather than disjoint documents. In this setting, we train on the RACE dataset (Lai et al., 2017), which consists of relatively short educational passages, and test on both RACE and the validation portion of QuALITY (Pang et al., 2022), which features much longer and more complex narratives. This setup allows us to assess whether document segmentation and aggregation via souping generalizes from short-form to long-form content. Our findings support earlier observations from the QuALITY paper that models trained on RACE can transfer effectively.

**Sparse Retrieval**   Finally, we evaluate sparse retrieval using the NIAH multikey-2 subset of the RULER dataset. In this setting, each document line contains a unique identifier paired with a corresponding value, and the model is tasked with memorizing all such mappings. At inference time, it receives a query specifying one of the identifiers and must output the correct value. The full input includes many such mappings, requiring the model to store a dense set of key-value pairs and retrieve the correct one after aggregation. This task challenges the model's ability to preserve fine-grained, instance-specific information across multiple independently encoded documents.

Table 3: EM (%) on the NIAH task for Mamba-2 8B models finetuned on 25K examples with either 4K (left) or 8K (right) sequence length. Models are evaluated across varying numbers of document segments and sequence lengths. Bold indicates the best score in each column. Gray cells mark results outperforming the respective concat-finetuned baseline (85.8 / 24.25 for 4K, 81.65 / 35.55 for 8K).

**Finetuned on 4K sequence length**

| Method | Train Segments | Test Segments | Test Seq. Length 4k | Test Seq. Length 8k |
|---|---|---|---|---|
| Concat | 1 | 1 | 85.8 | 24.25 |
| Soup w/ Average | 2 | 2 | 87.0 | 38.45 |
| | | 4 | 79.75 | 32.05 |
| | | 8 | 45.7 | 16.3 |
| | | 16 | 2.3 | 0.8 |
| | | 32 | 0.0 | 0.0 |
| | 4 | 2 | **88.6** | **40.7** |
| | | 4 | 86.8 | 38.7 |
| | | 8 | 76.55 | 29.3 |
| | | 16 | 29.05 | 11.45 |
| | | 32 | 0.9 | 0.5 |
| | 8 | 2 | 81.4 | 34.25 |
| | | 4 | 84.9 | 36.85 |
| | | 8 | 83.6 | 36.45 |
| | | 16 | 71.45 | 28.9 |
| | | 32 | 15.75 | 9.25 |

**Finetuned on 8K sequence length**

| Method | Train Segments | Test Segments | Test Seq. Length 4k | Test Seq. Length 8k |
|---|---|---|---|---|
| Concat | 1 | 1 | 81.65 | 35.55 |
| Soup w/ Average | 2 | 2 | **89.8** | **48.4** |
| | | 4 | 79.45 | 38.15 |
| | | 8 | 41.15 | 15.45 |
| | | 16 | 3.2 | 1.45 |
| | | 32 | 0.05 | 0.0 |
| | 4 | 2 | 88.5 | 46.55 |
| | | 4 | 86.5 | 43.85 |
| | | 8 | 75.65 | 34.6 |
| | | 16 | 23.8 | 11.5 |
| | | 32 | 0.4 | 0.55 |
| | 8 | 2 | 87.6 | 45.3 |
| | | 4 | 88.2 | 45.85 |
| | | 8 | 85.25 | 43.9 |
| | | 16 | 64.95 | 32.25 |
| | | 32 | 9.7 | 7.9 |

## 3.2 EXPERIMENTAL SETUP

**Training Configurations** All experiments use Mamba2-8B, as we observe that it consistently achieves stronger performance than the 2.7B model (see App. B for comparison), particularly in settings requiring multi-hop reasoning or generalization across segment length. We finetune our models using the AdamW optimizer with a cosine decay learning rate schedule. The 2.7B Mamba2 model is trained with a learning rate of $5 \times 10^{-5}$, while the 8B Mamba2 model uses $1 \times 10^{-5}$. For most experiments, we train for a single epoch. The only exception is the single-hop QA task, which, due to its smaller dataset size, is trained for three epochs to ensure convergence. For distributed training, we employ DeepSpeed ZeRO-2, which enables efficient extraction of the complete hidden states required for our souping mechanism. All other hyperparameters—including AdamW betas, gradient clipping values, warm-up details, and random seeds—along with specifics on the compute resources are detailed in Appendix A.4.

**Evaluation Metrics** We evaluate model performance using standard metrics for both extractive and multiple-choice question answering. For extractive QA tasks such as HotpotQA and RULER, we report **exact match (EM)**, which measures the percentage of predictions that exactly match any ground-truth answer span, and **F1 score**, which reflects the token-level overlap between the predicted and reference answers. For multiple-choice QA tasks, including RACE and QuALITY, we use **multiple-choice accuracy** as the evaluation metric. In this setting, the model outputs a distribution over the four answer options (A, B, C, D), and we compute the log-probability of each choice. The predicted answer corresponds to the option with the highest score, and accuracy is measured as the proportion of correctly selected answers.

## 3.3 BASELINES AND REFERENCE SETTING

We compare souping against several SSM baselines that represent different levels of supervision and context integration. First, we include a *pretrained* baseline: an off-the-shelf Mamba2 model used without any task-specific finetuning, which serves as a lower bound since the model has not been adapted to the QA tasks. Next, we consider a *QA-only finetuning* baseline trained only on $(q, a)$ pairs without access to any document context during training. Although this configuration does not support long-context integration, it shows how much performance is attributable to the context. Finally, we evaluate a *concat-based finetuning* baseline, where the model is trained end-to-end on full input sequences, i.e., $(d_1, \ldots, d_k, q, a)$, concatenated into a single flat input. This setting provides strong supervision by exposing the model to all relevant documents in a joint context window and serves as our primary reference point when assessing the performance of souping-based alternatives.

Table 4: Multiple-choice QA accuracy (%) for Mamba-2 8B finetuned on 25K RACE examples. Test is performed on RACE (5K) and QuALITY (2K) using answer selection based on maximum choice probability. Bolded values denote the highest score in each test. Gray cells indicate cases where soup-based finetuning outperforms concat-based finetuning.

| Method | Train Segments | Test Segments on RACE (5K) | | | | | | Test Segments on Quality (2K) | | | | | |
|---|---|---|---|---|---|---|---|---|---|---|---|---|---|
| | | 0 | 1 | 2 | 4 | 8 | 16 | 0 | 1 | 2 | 4 | 8 | 16 |
| Concat (QA-only) | 0 | 26.54 | – | – | – | – | – | 26.08 | – | – | – | – | – |
| Concat (With Context) | 1 | – | 56.01 | – | – | – | – | – | 32.69 | – | – | – | – |
| Soup w/ Average | 2 | – | – | 63.62 | 59.43 | 56.49 | 52.30 | – | – | 47.46 | 47.27 | 45.16 | 43.10 |
| | 4 | – | – | **66.02** | 63.54 | 60.58 | 57.31 | – | – | **51.15** | 50.05 | 48.08 | 45.69 |
| | 8 | – | – | 62.29 | 59.28 | 55.74 | 53.77 | – | – | 46.31 | 45.69 | 44.20 | 43.29 |

Our aim is to demonstrate that, with appropriate finetuning, souping-based models can outperform context-free baselines and approach the performance of the concat-based reference model.

**Justifying the Focus on SSMs vs. Transformers** A natural question is whether document souping can be extended to Transformer architectures for the sake of comparison. However, a direct application is conceptually challenging, as Transformers do not compress inputs into a fixed-size hidden state in the same manner as SSMs. The closest analogue to an SSM's state is the key-value (KV) cache. Crucially though, the KV cache's size is proportional to the input sequence length, making the aggregation of caches from documents of varying lengths non-trivial. To investigate empirically, we designed an experiment to test a cache souping approach. To address the size mismatch, all documents were padded to a uniform length before being encoded independently. We implemented a cache souping variant for LLaMA 3 8B on HotpotQA, averaging the KV caches from ten documents (see Appendix A.5.1 for full experimental details). The experiment confirmed this approach is ineffective for Transformers: while standard finetuning on concatenated documents achieved 48.50 EM and 62.81 F1, cache souping yielded only 0.53 EM and 8.18 F1 given the same amount of data. This significant performance degradation supports our hypothesis that the fixed-size, recurrent state of SSMs is a critical architectural feature for this style of modular recombination. Consequently, we focus our main experiments on SSM baselines.

## 4 RESULTS

### 4.1 FINDING 1: SSM FINETUNING UNLOCKS DOCUMENT SOUPABILITY

Finetuning is crucial for unlocking soupability in pretrained SSMs. We systematically evaluate three training strategies on HotpotQA, with results presented in Table 1. Our findings demonstrate a clear progression in performance as more components of the model are adapted to the task of interpreting souped states. First, we test an **out-of-the-box** approach by merging hidden states from a pretrained Mamba2 model without any finetuning. As shown in the top rows of Table 1, this yields poor performance (e.g., 2.6 EM on 5 documents), confirming that the model cannot interpret pooled representations naively. Second, we evaluate a **decoder-only finetuning** strategy. This results in a substantial performance gain (38.8 EM on 5 documents), demonstrating that the decoder can learn to interpret the fixed representations produced by an unadapted encoder. Third, **full encoder-decoder finetuning** trains both components jointly. This approach consistently achieves the strongest results (47.8 EM on 5 documents), outperforming the decoder-only method. This indicates that optimal performance is achieved when the encoder is also trained to produce more effectively mergeable states. These findings confirm that SSMs can be trained end-to-end to support powerful, modular reasoning through state pooling.

### 4.2 FINDING 2: AVERAGING DOCUMENT REPRESENTATIONS IMPROVES SOUPABILITY

We evaluate several strategies for aggregating per-layer hidden states, including elementwise **average**, **sum**, and **maximum**, with optional **unit normalization**. The results, detailed in Table 1, demonstrate that *no-norm averaging* is consistently the most effective and stable approach. This method achieves a top score of 47.8 EM and 61.3 F1 on 5 documents. We hypothesize that this stability stems from the ability to aggregate information across documents while inherently keeping the magnitude

Table 5: EM / F1 scores on RULER QA_1 for Mamba2-8B trained and evaluated on 4k sequence length. Gray cells indicate performance exceeding the concat-finetuned test results (EM 54.81 / F1 71.90), and bold marks the highest test result of the task. Training on more segments improves generalization to higher evaluation segment counts.

| Method | Train Segments | Test Segments | | | | |
|---|---|---|---|---|---|---|
| | | 1 | 2 | 5 | 10 | 20 |
| Concat | 1 | 54.81 / 71.90 | – | – | – | – |
| Soup w/ Average | 2 | – | **58.38 / 74.05** | 34.89 / 49.04 | 13.19 / 23.40 | 12.36 / 22.45 |
| | 5 | – | **58.38** / 73.89 | 57.28 / 72.09 | 35.71 / 50.80 | 14.97 / 26.25 |
| | 10 | – | 21.02 / 31.41 | 13.87 / 22.36 | 52.75 / 68.68 | 43.82 / 58.99 |
| | 20 | – | 28.71 / 41.44 | 28.85 / 42.93 | 45.60 / 61.78 | 44.37 / 60.85 |

of the resulting state vector from growing with the number of inputs. In contrast, other methods are less robust, particularly as the number of documents grows. For example, *summation without normalization* degrades sharply from 44.2 EM on 5 documents to just 25.0 EM on 10 documents. We attribute this to the unbounded growth in activation magnitude. While applying normalization after summation mitigates this collapse, recovering the score to 34.7 EM, it still underperforms the simple averaging result of 38.7 EM on 10 documents. Interestingly, normalization prior to summation performs poorly, likely because it destroys relative magnitude differences across documents that are helpful when using sum. Given these findings, we conclude that averaging without normalization offers the best trade-off between simplicity, stability, and generalization. We therefore adopt it as our default pooling strategy for all subsequent experiments.

### 4.3 FINDING 3: DOCUMENT SOUPING IS SCALABLE AND GENERALIZES TO WIDE CONTEXTS

Our analysis reveals two key principles that govern how document souping generalizes across different context sizes, demonstrated consistently across a range of diverse tasks (Tables 2, 3, 4, and 5). First, **performance is usually maximized when the inference-time soup size aligns with the training configuration**. This trend is clearly visible in our long-document QA results (Table 4). For instance, on the QuALITY benchmark, a model trained on 4 document segments achieves 50.05% accuracy when tested on 4 segments, and as the number of test segments increases to 8 and 16, it gradually declines to 48.08% and 45.69%, respectively. Second, **models finetuned on a greater number of segments exhibit more robust generalization**. This is demonstrated in our Needle-in-a-Haystack experiments (Table 3). A model trained on only 2 segments with 4k sequence length fails completely when tested on 16 segments with the same sequence length (2.3 EM), while a model trained on 8 segments maintains a strong EM score of 71.45% when tested on 16. This shows that exposing the model to wider contexts during finetuning is key for robust generalization. To stress-test the limits of this generalization, we conducted scalability experiments up to 256 documents (Appendix B.1, Table 8). The findings are twofold: a model trained on 64 documents fails to extrapolate to 256, with its score collapsing from **39.75 to 0.10 EM**. However, by continuing to finetune this checkpoint on 128 documents, the model becomes robust at these wider contexts, achieving a strong **37.25 EM** on 256 documents. This result proves that our method is not limited by an architectural bottleneck but can be effectively adapted to handle vast contexts. This capability is particularly relevant for applications like retrieval-augmented generation (RAG), where a retriever provides a variable but bounded number of documents (e.g., 20-100) for synthesis. Our results show that document souping provides a practical and highly scalable paradigm for this task. By aligning the finetuning strategy with the expected operational parameters of the RAG system, our method enables robust reasoning over large, dynamic document sets.

### 4.4 FINDING 4: SOUPING AMPLIFIES CACHING BENEFITS FOR EFFICIENT INFERENCE

Document souping dramatically improves inference efficiency by enabling a modular caching strategy. Our latency benchmarks, presented in Table 6, demonstrate the substantial speedups achieved by this approach, which excludes the one-time, offline cost of encoding documents. While gains are modest for short sequences, the benefit grows significantly with context length. For a 32k token document, for example, loading the cached state and processing the query takes only **65.8 ms**, compared to the 963.2 ms required for the standard concatenation method. This speedup is amplified by the modularity

of our approach. Because each document's state is cached independently, **any arbitrary subset of documents can be composed on-the-fly** at inference time without re-computation. This provides a crucial advantage over monolithic caching schemes, where changing even a single document in a retrieved set would require reprocessing the entire context. This flexibility is particularly powerful for retrieval-augmented generation (RAG) systems, where different queries retrieve unique combinations of documents, making our method a highly efficient solution for dynamic, large-scale reasoning.

## 5 RELATED WORK

**Long-Context Sequence Modeling.** Transformers (Vaswani et al., 2017) remain the dominant architecture for modeling long-range dependencies, but their $\mathcal{O}(L^2)$ attention cost and $\mathcal{O}(L)$ memory growth in the KV cache hinder scalability. Many works explore architectural modifications to overcome this, including sparse attention (Child et al., 2019; Beltagy et al., 2020; Zaheer et al., 2020), linearized attention (Katharopoulos et al., 2020), and chunked processing with memory compression (Dai et al., 2019; Wu et al., 2022; Xiao et al., 2024). Retrieval-augmented generation methods, such as RAG (Lewis et al., 2020) offloads long-context memory to external sources.

Structured State Space Models (SSMs) offer an alternative path with fundamentally different scaling characteristics. Models like S4 (Gu et al., 2021; 2022) introduced linear-time computation via structured recurrences. Mamba (Gu & Dao, 2024) builds on this with input-dependent gating, while Mamba2 (Dao & Gu, 2024) unifies SSMs and attention through structured state space duality, yielding substantial speedups on long sequences with strong downstream performance. Recent work further demonstrates the utility of Mamba-based models for dense and retrieval-augmented tasks, including dense passage ranking (Zhang et al., 2024) and long-document retrieval in RAG (Cao et al., 2025).

**Model and Representation Souping.** Model souping refers to merging parameters across finetuned checkpoints to improve robustness and generalization without retraining (Wortsman et al., 2022; Tang et al., 2024). More recently, several works have explored souping at the level of internal representations rather than weights. State Soup (Pióro et al., 2024) stores in-context *skills* as recurrent states and shows that they can be retrieved and linearly mixed for task transfer. PICASO (Liu et al., 2025) derives a permutation-invariant operator for composing SSM states as an approximation to full concatenation. We extend these works by focusing on souping hidden states corresponding to disjoint document chunks, and study when such aggregated representations can support *factual*, multi-document reasoning on real downstream benchmarks. Our work characterizes both the regimes in which document-level state souping is effective and the limitations of souping strategies.

**Parallel and Distributed Ingestion.** Efficient ingestion of long contexts has been addressed through sparse or structured attention mechanisms (Child et al., 2019; Zaheer et al., 2020), memory compression across chunks (Dai et al., 2019; Wu et al., 2022), and retrieval-based pipelines (Lewis et al., 2020). For SSMs, constant-memory recurrence and linear compute make them naturally suited to chunk-wise streaming. However, to our knowledge, no prior work has explicitly studied post-hoc merging of hidden states across chunks for joint reasoning.

**RNN-Inspired State Mixing.** Recurrent models have long supported state-passing across time steps, and early works explored implicit mixing of information through recurrent transitions (Grossberg, 2013). Our approach differs in that it investigates *explicit* aggregation of intermediate hidden states produced in parallel, rather than temporal chaining of recurrent updates.

**Key/Value Cache Merging.** Recent works explored compressing text by adaptive merging KV caches (Wang et al., 2024), and cache eviction policies (Zhang et al., 2023). While related in spirit, these approaches focus on attention-based caches, whereas our work targets SSM hidden states.

**Token Merging in Vision Transformers.** In the vision domain, token merging has been studied as a means of compressing long sequences. TCFormer (Zeng et al., 2022) clusters tokens across space for progressive downsampling. Other work merges tokens dynamically based on content (Bolya et al., 2022; Kim et al., 2024), or selects value/key pairs to retain or evict during inference (Wan et al., 2024). These ideas echo our motivation to aggregate compact, composable representations, though in our case applied to document states within state space models.

| Doc SeqLen | Concat Doc + Ques (ms) | Cache States + Ques (ms) | Speedup |
|---|---|---|---|
| 2048 | 60.7 | 59.7 | 1.0× |
| 4096 | 164.2 | 62.3 | 2.6× |
| 8192 | 235.9 | 59.5 | 4.0× |
| 16384 | 469.7 | 60.9 | 7.7× |
| 32768 | 963.2 | 65.8 | 14.6× |

Table 6: Inference latency with cached states. Souping amplifies caching by allowing **any subset** of documents to be dynamically composed without re-computation. The initial encoding cost is excluded.

## 6 CONCLUSION

We study document souping, a method for merging hidden states across independently encoded documents in structured state space models (SSMs), and analyze its behavior on downstream tasks. By aggregating layer-wise representations through simple commutative operations, souping enables modular, parallel ingestion of long-context inputs while supporting accurate downstream inference. Our experiments demonstrate that Mamba2 models trained with souping are capable of multi-hop reasoning, sparse retrieval, and generalization across document and segment scales, often matching or outperforming traditional concat-based finetuning. Overall, our findings highlight state souping as a lightweight and effective strategy for long-context reasoning in SSMs, and we hope it provides a foundation for broader adoption and further exploration.

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

# A  IMPLEMENTATION DETAILS

## A.1  DATA FORMATTING

To support both standard fine-tuning and our proposed souping method, we define two input formats:

- **Concat-data**: the input $x$ is formed by linearly concatenating $k$ documents, the question $q$, the answer $a$, and an end-of-sequence token $\langle \text{eos} \rangle$:

$$x = d_1 \oplus d_2 \oplus \cdots \oplus d_k \oplus q \oplus a \oplus \langle \text{eos} \rangle.$$

  This aligns with conventional LM fine-tuning, where the model attends jointly over the entire sequence.

- **Souping-data**: identical to concat-data, except that after each document $d_i$ we insert a special separator token $\langle \text{DOC\_SEP} \rangle$. Formally,

$$x = d_1 \oplus \langle \text{DOC\_SEP} \rangle \oplus d_2 \oplus \langle \text{DOC\_SEP} \rangle \oplus \cdots \oplus d_k \oplus \langle \text{DOC\_SEP} \rangle \oplus q \oplus a \oplus \langle \text{eos} \rangle.$$

  During preprocessing, we split on $\langle \text{DOC\_SEP} \rangle$ to isolate each document chunk for parallel encoding.

**Notation.** We use $\oplus$ to denote string concatenation, $\langle \text{eos} \rangle$ to mark end-of-sequence, and $\langle \text{DOC\_SEP} \rangle$ to separate documents. Algorithm 1 summarizes the procedure for constructing $x$ under both modes.

---

**Algorithm 1** Input Formatting for Concat- vs. Souping-data

---

**Require:** Documents $\{d_1, \ldots, d_k\}$, question $q$, answer $a$, flag soup $\in \{\text{true}, \text{false}\}$
**Ensure:** Formatted input $x$
1: $x \leftarrow$ empty string
2: **for** $i = 1$ to $k$ **do**
3:     **if** soup **then**
4:         $x \leftarrow x \oplus d_i \oplus \langle \text{DOC\_SEP} \rangle$
5:     **else**
6:         $x \leftarrow x \oplus d_i \oplus$ `" "`
7:     **end if**
8: **end for**
9: $x \leftarrow x \oplus q \oplus$ `" "` $\oplus a \oplus \langle \text{eos} \rangle$

---

## A.2  SOUPING RECIPE

### A.2.1  WITHOUT ACTIVATION CHECKPOINTING

Given a souping-data sequence $x$, we split at each $\langle \text{DOC\_SEP} \rangle$, producing $k + 1$ segments:

$$\{s_1, \ldots, s_k, s_{k+1}\},$$

where $s_i = d_i$ for $i \leq k$ and $s_{k+1}$ contains $(q, a, \langle \text{eos} \rangle)$. We then form:

1. **Doc-batch formation.** Stack the $k$ context segments $\{s_1, \ldots, s_k\}$ along the batch dimension, yielding a tensor of shape $(B \times k, T, D)$, where $B$ is the original mini-batch size, $T$ is the (left-padded) segment length, and D is embedding dimension.

2. **Parallel encoding.** Pass the Doc-batch through the encoder in parallel to obtain per-document hidden states $\{h_i\}_{i=1}^{B \times k}$ for each batch element.

3. **Aggregation.** Merge documents $\{h_i\}$ for each sequence of the mini-batch via the chosen souping strategy (average, sum, or max) to produce a single *souped* state $\{h_{\text{soup\_j}}\}_{j=1}^{B}$ per batch element.

4. **QA-batch processing.** Take the QA segment $s_{k+1}$, right-pad it to length $T$ to form a QA-batch of shape $(B, T, D)$. We feed the QA-batch along with $h_{\text{soup}}$ injected at every layer—through the decoder. The souped state remains constant until decoding begins.

5. **Loss computation.** Map decoder outputs back to token positions, convert to logits, and compute cross-entropy loss over the answer tokens.

### A.2.2 WITH ACTIVATION CHECKPOINTING

To support training with more documents and larger model sizes without running out of memory, we employ document-level activation checkpointing. Unlike the parallel encoding scheme in Section A.2.1, which processes all context documents of a mini-batch simultaneously, we encode *one document per mini-batch at a time* under checkpointing.

Specifically, we iterate over the $k$ context segments for each batch and perform a forward pass for a single document per step, using PyTorch's gradient checkpointing to trade off compute for memory. During backpropagation, each document's encoder activations are recomputed on-the-fly. This significantly reduces peak memory usage, enabling us to scale to more documents (higher $k$), longer input sequences, and larger model weights.

While this sequential encoding incurs additional compute overhead, it offers a practical trade-off to unlock training regimes otherwise inaccessible due to memory constraints. If sufficient GPU memory is available (e.g., large HBM capacity), the overhead of activation checkpointing can be amortized by increasing the batch size—sometimes even resulting in faster end-to-end training than without checkpointing, due to improved throughput and parallelism.

### A.3 PSEUDO-CODE

Algorithm 2 describes the inference-time procedure for corpus encoding via document souping. Each document $d_i$ is processed independently by a shared SSM encoder to produce a set of per-layer hidden states $\{h_i^{(1)}, \ldots, h_i^{(L)}\}$. Optionally, each hidden state can be normalized (e.g., to unit norm) before being appended to a layer-specific collection. After all documents are encoded, hidden states are aggregated across documents at each layer using a commutative pooling operation such as averaging, summation, or max. The resulting pooled states $\{h_{\text{soup}}^{(1)}, \ldots, h_{\text{soup}}^{(L)}\}$ form a compact, merged representation of the document set, which is then used to condition the decoder when answering a query $q$. The document encoding can be performed offline and cached in advance, enabling efficient reuse across queries.

---

**Algorithm 2** Corpus Encoding via Document Souping (Inference Only)

---

**Require:** Document set $\{d_1, \ldots, d_k\}$, query $q$
**Ensure:** Predicted answer $\widehat{y}$
 1: Initialize empty list of layerwise states $\{H^{(l)}\}_{l=1}^{L}$
 2: **for** each document $d_i$ **do**
 3:     Compute hidden states: $\{h_i^{(1)}, \ldots, h_i^{(L)}\} \leftarrow \text{SSM}_\theta(d_i)$
 4:     **for** $l = 1$ to $L$ **do**
 5:         Optionally normalize: $\tilde{h}_i^{(l)} \leftarrow h_i^{(l)}/\|h_i^{(l)}\|$
 6:         Append $\tilde{h}_i^{(l)}$ to $H^{(l)}$
 7:     **end for**
 8: **end for**
 9: **for** $l = 1$ to $L$ **do**
10:     Pool document states: $h_{\text{soup}}^{(l)} \leftarrow \text{pool}(H^{(l)})$
11:     Optionally normalize: $h_{soup}^{(l)} \leftarrow h_{soup}^{(l)}/\|h_{soup}^{(l)}\|$
12: **end for**
13: Predict: $\widehat{y} \leftarrow \text{SSM}_\theta\left(q \mid \{h_{\text{soup}}^{(l)}\}_{l=1}^{L}\right)$

---

### A.4 IMPLEMENTATION AND TRAINING DETAILS

This section provides a comprehensive overview of the hyperparameters, software, and hardware used in our experiments.

Table 7: HotpotQA results for Mamba-2 2.7B with 2 gold documents and $n-2$ distractors. Cells show **EM / F1**. Underline marks models tested on the same docs they are trained on.

| Model | Evaluation: 2 **gold** + $(n-2)$ **distractors** | | | | | | | | | |
|---|---|---|---|---|---|---|---|---|---|---|
| | $n=0^\dagger$ | 2 | 3 | 4 | 5 | 6 | 7 | 8 | 9 | 10 |
| **Pretrained, no finetune** | | | | | | | | | | |
| *Concat* | 3.3 / 7.8 | 8.4 / 22.7 | 5.8 / 18.6 | 5.2 / 17.1 | 3.5 / 14.0 | 3.2 / 12.9 | 2.8 / 11.8 | 2.2 / 11.6 | 2.2 / 11.0 | 2.3 / 10.4 |
| **Finetuned on $n=0$ $gold + 0$ $distractors$ (QA-only)** | | | | | | | | | | |
| Concat | 12.7 / 19.4 | 40.9 / 54.2 | 36.7 / 49.1 | 32.4 / 44.7 | 30.4 / 41.8 | 28.0 / 39.2 | 26.2 / 36.9 | 25.4 / 35.5 | 23.6 / 33.7 | 22.3 / 32.2 |
| **Finetuned on $n=2$ $gold + 0$ $distractors$ documents** | | | | | | | | | | |
| Concat | – | 50.8 / 65.2 | 46.0 / 59.2 | 41.7 / 54.5 | 37.7 / 49.7 | 35.9 / 47.5 | 33.8 / 44.6 | 30.1 / 40.9 | 28.4 / 38.4 | 26.6 / 36.9 |
| Soup w/ Average | – | 47.4 / 61.0 | 39.9 / 52.7 | 33.0 / 45.5 | 28.8 / 40.4 | 25.5 / 36.8 | 23.8 / 34.2 | 21.9 / 31.8 | 19.7 / 29.2 | 19.0 / 28.2 |
| **Finetuned on $n=2$ $gold + 3$ $distractors$ documents** | | | | | | | | | | |
| Concat | – | 48.8 / 63.5 | 46.3 / 60.2 | 43.1 / 56.8 | 42.3 / 55.2 | 40.6 / 53.6 | 37.1 / 49.6 | 35.3 / 48.0 | 35.2 / 47.6 | 33.7 / 45.6 |
| Soup w/ Average | – | 49.1 / 62.9 | 44.8 / 58.3 | 41.5 / 54.4 | 38.4 / 51.0 | 35.7 / 48.2 | 33.7 / 45.5 | 32.7 / 43.8 | 30.1 / 41.3 | 28.6 / 39.3 |
| **Finetuned on $n=2$ $gold + 5$ $distractors$ documents** | | | | | | | | | | |
| Concat | – | 50.4 / 64.4 | 47.0 / 60.9 | 42.7 / 56.4 | 41.8 / 55.0 | 40.1 / 52.8 | 38.1 / 50.3 | 37.1 / 49.4 | 36.9 / 48.8 | 34.8 / 46.7 |
| Soup w/ Average | – | 48.8 / 63.0 | 44.2 / 57.8 | 40.3 / 53.5 | 37.7 / 50.5 | 36.0 / 48.2 | 33.6 / 45.6 | 32.0 / 43.7 | 30.1 / 41.2 | 28.5 / 39.5 |
| **Finetuned on $n=2$ $gold + 8$ $distractors$ documents** | | | | | | | | | | |
| Concat | – | 49.7 / 63.7 | 45.2 / 58.8 | 43.6 / 56.7 | 41.7 / 54.1 | 40.0 / 53.1 | 36.8 / 48.8 | 36.4 / 48.4 | 35.3 / 47.4 | 33.9 / 45.7 |
| Soup w/ Average | – | 49.5 / 63.2 | 45.8 / 58.8 | 42.6 / 55.1 | 39.2 / 51.2 | 37.6 / 48.9 | 36.1 / 47.1 | 34.4 / 45.1 | 32.5 / 43.2 | 31.2 / 41.6 |

$^\dagger$ $n=0$ corresponds to a no-context setting where the model receives only the query (i.e., 0 gold documents and 0 distractors).

### A.4.1 TRAINING HYPERPARAMETERS AND COMPUTE RESOURCES

To ensure reproducibility, all experiments were conducted with the random seed fixed to 42 and PyTorch's deterministic mode enabled (`deterministic=True`). Our optimization strategy is based on the setup from prior work on Mamba. The specific hyperparameters are as follows:

- **Optimizer**: AdamW

- **AdamW Betas**: We use $\beta_1 = 0.9$ and $\beta_2 = 0.95$.

- **Learning Rate Schedule**: A cosine decay schedule was used with a linear warm-up for the first 10% of training steps.

- **Gradient Clipping**: Gradients were clipped at a maximum norm of 1.0.

- **Gradient Accumulation**: We used 120 accumulation steps for most configurations. For the Mamba2-2.7B model on HotpotQA, this was set to 128 steps.

Experiments were conducted on clusters equipped with NVIDIA H100 and H200 GPUs.

### A.5 DECODING HYPERPARAMETERS

We optimized decoding strategies for both CONCAT and SOUP /W AVERAGE using a subset of HotpotQA comprising 30K training examples and 3K validation examples, each containing 5 documents (2 gold, 3 distractors). A grid search was performed over temperature $\{0.3, 0.5, 0.7, 0.9\}$, top-$p$ $\{0.5, 0.7, 0.9, 0.95\}$, and top-$k$ $\{5, 10, 20, 30, 40\}$.

SOUP achieved best validation performance with temperature $= 0.3$, top-$p = 0.5$, and top-$k = 20$, while CONCAT performed optimally with temperature $= 0.5$, top-$p = 0.95$, and top-$k = 30$. All subsequent experiments adopted these respective configurations for each method.

### A.5.1 TRANSFORMER CACHE SOUPING EXPERIMENTAL SETUP

To test the viability of extending our souping method to Transformer architectures, we conducted a comparative experiment using LLaMA3-8B Instruct. We finetuned the model for one epoch on 10,000 examples from the HotpotQA dataset, where each example consisted of two gold documents and eight distractor documents. We compared two distinct fine-tuning methodologies:

- **Normal Fine-tuning (Baseline):** The model was fine-tuned on the simple concatenation of all 10 documents and the question. We used a learning rate of $1 \times 10^{-5}$ for this setup.

- **KV Cache Souping Fine-tuning:** To handle variable-length inputs, all documents were left-padded to match the longest document in the batch. Each document was then processed independently to generate its key-value (KV) cache. The resulting caches were averaged to create a single, souped KV cache, which was then used alongside the question for downstream reasoning. A learning rate of $1 \times 10^{-6}$ was used for this approach.

### A.5.2 COMPUTE RESOURCES

Experiments were conducted using a combination of on-premise and cloud nodes, equipped with high-memory GPUs and multi-core CPUs. All runs used PyTorch 2.6 with CUDA 12.4.

- **Node A (On-premise):** 8× NVIDIA L40S (48GB) with AMD EPYC 7282 CPU
- **Node B (On-premise):** 8× NVIDIA H100 (80GB) with Intel Xeon Platinum 8480+ CPU
- **Node C (Cloud):** On-demand NVIDIA H200 (144GB) with AMD EPYC 9654 CPU

Nodes A and B were used for model development, ablations, and full training runs. Node C enabled rapid parallel experimentation.

Training time varies with model size, input sequence length, number of documents per example, and total steps. For instance, fine-tuning an 8B model on HotpotQA (30K examples, 5 documents per example) requires approximately 3 hours on a single H200 GPU with activation checkpointing enabled.

Multi-document souping increases memory overhead. The following configurations represent minimum requirements for training on HotpotQA with 5-document inputs and small batch sizes:

- **2.7B model:** $\geq$ 1× 80GB or 2× 48GB GPUs
- **8B model:** $\geq$ 1× 144GB, 2× 80GB, or 4× 48GB GPUs

Training with longer input sequences or larger segments proportionally increases memory demands, requiring additional GPUs or higher-capacity HBM.

## B ADDITIONAL RESULTS AND ANALYSIS

### B.1 SCALABILITY OF SOUP SIZES

To further probe the limits of scalability, we conducted experiments up to 256 documents. To stay consistent with previous results for fair comparison, we evenly splitted the original 8192 sequence length NIAH tasks into 64, 128, and 256 segments. The full results of this scalability analysis are presented in Table 8. We first finetuned a Mamba2-8B model on the NIAH dataset using 64 segments for 25,000 samples. While this model generalized well to nearby segment counts, its performance dropped sharply when tested on 128 or 256 segments. However, by continuing to finetune this checkpoint on 128 segments for 8,400 samples, the model's performance at these higher counts improved dramatically. Compared to concatenating finetuning baseline's result 35.55 in Table 3, the continued checkpoint reaches a score of 39.00 at 128 segments and 37.25 at 256 segments, outperforming the baseline while maintaining robustness across smaller soup sizes. This two-stage process demonstrates that while extrapolation is challenging, the model can be effectively trained to handle very large soup sizes when provided with appropriate data.

| Test Segments | Finetuned on 64 Segments | Continued on 128 Segments |
|---|---|---|
| 2 | 28.65 | 23.65 |
| 4 | 34.40 | 26.60 |
| 8 | 37.95 | 27.70 |
| 16 | 40.05 | 30.25 |
| 32 | 41.65 | 32.30 |
| 64 | 39.75 | 37.80 |
| 128 | 15.15 | 39.00 |
| 256 | 0.10 | 37.25 |

Table 8: Scalability results (EM %) on the NIAH dataset for a Mamba2-8B model with an 8k sequence length. We compare a model finetuned on 64 segments against a model that was subsequently trained further on 128 segments. Continued training significantly improves performance on larger soup sizes, demonstrating the model's capacity to adapt to wider contexts.

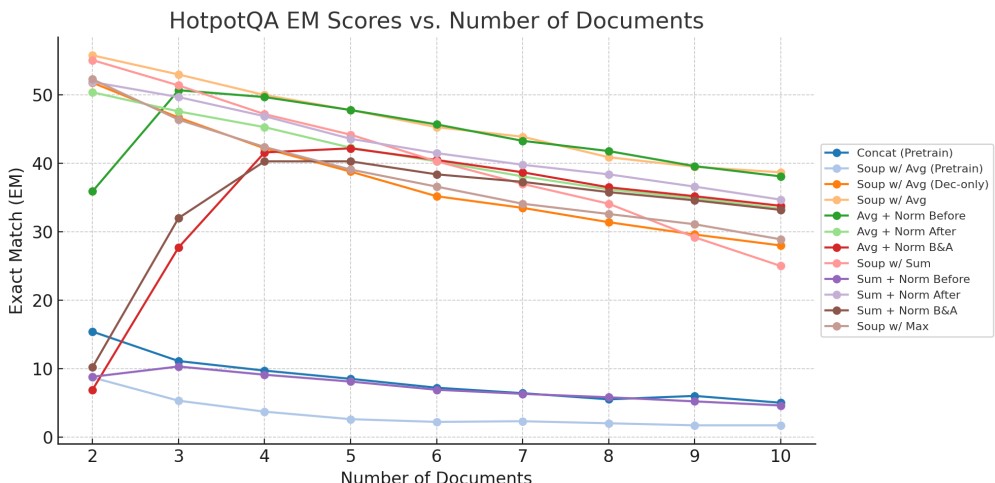

Figure 2: Exact Match (EM) scores on HotpotQA for Mamba2-8B evaluated across increasing numbers of input documents. Each line represents a model trained on 5 documents (2 gold + 3 distractors) with a different pooling and normalization configuration. Soup w/ Average consistently remains robust across all tested document sizes compared to other configurations.

## B.2 HOTPOTQA EXTENDED ANALYSIS

### B.2.1 MAMBA2-2.7B RESULTS

Table 7 shows HotpotQA results for Mamba2-2.7B across a range of training and evaluation soup sizes. We observe that performance improves consistently with fine-tuning and scales with the number of documents seen during training. However, compared to the 8B model (Table 2), the 2.7B model shows greater sensitivity to soup size mismatch and a more pronounced performance drop at larger test-time document counts. This supports our finding that larger models generalize more robustly across soup sizes and better tolerate distribution shift during inference.

### B.2.2 MAMBA2-8B RESULTS

Table 9 expands on the main paper's Table 1 by providing a more detailed view of HotpotQA results for the Mamba2-8B model trained on 5 documents (2 gold + 3 distractors). At inference time, we evaluate the same checkpoint across a range of input sizes, including a no-context setting (query-only) and settings with 2 to 10 documents, where each includes 2 gold documents and $n-2$ distractors.

The table includes comparisons across several pooling operators (average, sum, max) and normalization variants, and includes pretrained, decoder-only, and full encoder-decoder fine-tuning settings. The results show that while all models degrade slightly as the number of input documents increases, full encoder-decoder fine-tuning with no-norm averaging consistently yields the best performance across soup sizes. Overall, these results highlight the robustness of soup-based representations under increasing input width and reinforce the conclusions drawn in the main text.

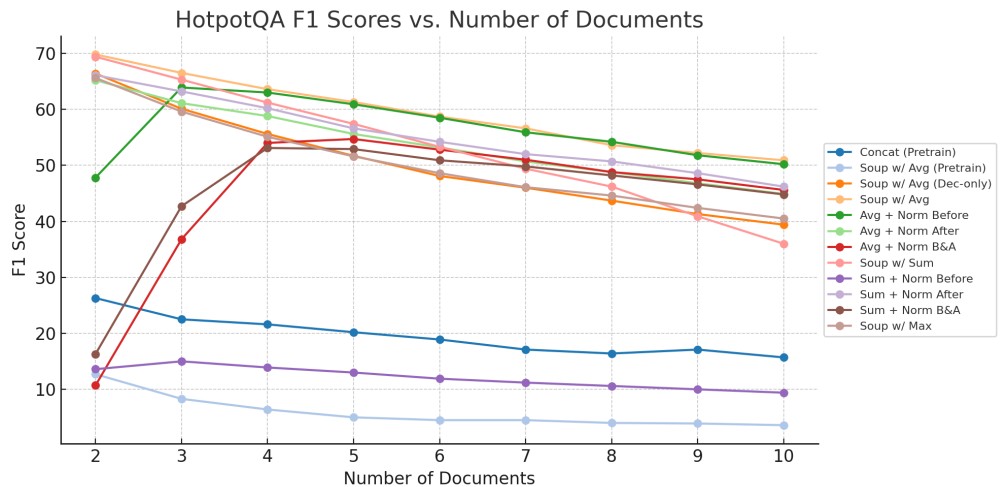

Figure 3: F1 scores on HotpotQA for Mamba2-8B using the same experimental setup as Figure 2.

Table 9: HotpotQA performance (Exact Match / F1) for Mamba-2 8B trained on 5 documents (2 gold + 3 distractors) and tested on $n$ documents, with 2 gold and $(n - 2)$ distractors. We compare pretrained models, decoder-only fine-tuning, and encoder-decoder fine-tuning, under different pooling operators (Avg, Sum, Max) and normalization settings. Underline marks models tested on the same docs they are trained on.

| Method | Test on 2 gold + $(n - 2)$ distractors | | | | | | | | |
|---|---|---|---|---|---|---|---|---|---|
| | 2 | 3 | 4 | 5 | 6 | 7 | 8 | 9 | 10 |
| **Pretrained 8B (No Finetune)** | | | | | | | | | |
| Concat | 15.4 / 26.3 | 11.1 / 22.5 | 9.7 / 21.6 | 8.5 / 20.2 | 7.2 / 18.9 | 6.4 / 17.1 | 5.5 / 16.4 | 6.0 / 17.1 | 5.0 / 15.7 |
| Soup w/ Average | 8.7 / 12.7 | 5.3 / 8.3 | 3.7 / 6.4 | 2.6 / 5.0 | 2.2 / 4.5 | 2.3 / 4.5 | 2.0 / 4.0 | 1.7 / 3.9 | 1.7 / 3.6 |
| **Decoder-Only Finetuned 8B** | | | | | | | | | |
| Soup w/ Average | 51.8 / 66.4 | 46.7 / 60.1 | 42.2 / 55.6 | 38.8 / 51.7 | 35.2 / 48.1 | 33.5 / 46.0 | 31.4 / 43.7 | 29.6 / 41.3 | 28.0 / 39.4 |
| **Encoder-Decoder Finetuned 8B** | | | | | | | | | |
| **Full Finetuned - Average Pooling With & Without Norms** | | | | | | | | | |
| Soup w/ Average | 55.8 / 69.8 | 53.0 / 66.5 | 50.0 / 63.6 | 47.8 / 61.3 | 45.3 / 58.7 | 43.9 / 56.6 | 40.9 / 53.6 | 39.5 / 52.2 | 38.7 / 50.9 |
| Average + Norm Before | 35.9 / 47.8 | 50.7 / 63.9 | 49.7 / 63.0 | 47.8 / 60.9 | 45.7 / 58.5 | 43.3 / 55.9 | 41.8 / 54.2 | 39.6 / 51.8 | 38.1 / 50.2 |
| Average + Norm After | 50.4 / 65.2 | 47.6 / 61.1 | 45.3 / 58.8 | 42.3 / 55.6 | 40.3 / 53.2 | 38.1 / 50.7 | 36.2 / 48.8 | 34.9 / 46.8 | 33.4 / 44.9 |
| Average + Norm Before & After | 6.9 / 10.7 | 27.7 / 36.8 | 41.6 / 54.0 | 42.2 / 54.7 | 40.5 / 52.8 | 38.7 / 51.0 | 36.5 / 48.8 | 35.2 / 47.5 | 33.8 / 45.6 |
| **Full Finetuned - Summation Pooling With & Without Norms** | | | | | | | | | |
| Soup w/ Sum | 55.1 / 69.4 | 51.4 / 65.3 | 47.2 / 61.2 | 44.2 / 57.4 | 40.3 / 53.3 | 37.0 / 49.4 | 34.1 / 46.2 | 29.2 / 40.9 | 25.0 / 36.0 |
| Sum + Norm Before | 8.8 / 13.6 | 10.3 / 15.0 | 9.1 / 13.9 | 8.1 / 13.0 | 6.9 / 11.9 | 6.3 / 11.2 | 5.8 / 10.6 | 5.2 / 10.0 | 4.6 / 9.4 |
| Sum + Norm After | 51.9 / 66.1 | 49.7 / 63.2 | 46.9 / 60.2 | 43.6 / 56.6 | 41.5 / 54.2 | 39.8 / 52.0 | 38.4 / 50.7 | 36.6 / 48.6 | 34.7 / 46.2 |
| Sum + Norm Before & After | 10.2 / 16.3 | 32.0 / 42.7 | 40.3 / 53.1 | 40.3 / 52.9 | 38.4 / 50.9 | 37.3 / 49.8 | 35.8 / 48.2 | 34.6 / 46.6 | 33.2 / 44.8 |
| **Full Finetuned - Max Pooling Without Norms** | | | | | | | | | |
| Soup w/ Max | 52.3 / 65.6 | 46.4 / 59.6 | 42.4 / 55.1 | 39.1 / 51.6 | 36.6 / 48.6 | 34.1 / 46.1 | 32.6 / 44.6 | 31.1 / 42.4 | 28.9 / 40.5 |

To visualize these trends, Figure 2 and Figure 3 plot EM and F1 scores, respectively, across different input sizes. These plots confirm that while pretrained and decoder-only models degrade rapidly, full finetuning with average pooling remains more robust. The visualizations emphasize how different configurations respond to increasing distractor load, further illustrating the stability of soup representations under input variation.

Table 10: Accuracy on EM (%) on the NIAH task for Mamba2-2.7B models trained with 4K (left) and 8K (right) sequence lengths for 25K examples. Bold indicates the best result in each column. Gray cells indicate accuracy exceeding the Concat-finetuned baseline (84.4 / 36.1 for 4K, 78.3 / 43.05 for 8K).

| | Finetuned on 4K Sequence Length | | | | | Finetuned on 8K Sequence Length | | | |
|---|---|---|---|---|---|---|---|---|---|
| Method | Train Segments | Test Segments | Test Seq. Length | | Method | Train Segments | Test Segments | Test Seq. Length | |
| | | | 4k | 8k | | | | 4k | 8k |
| Concat | 1 | 1 | 84.4 | 36.1 | Concat | 1 | 1 | 78.3 | 43.05 |
| Soup w/ Avg. | 2 | 2 | 84.4 | 33.9 | Soup w/ Avg. | 2 | 2 | 84.4 | 59.2 |
| | | 4 | 71.7 | 35.5 | | | 4 | 71.6 | 47.8 |
| | | 8 | 26.2 | 12.9 | | | 8 | 28.5 | 15.1 |
| | | 16 | 0.3 | 0.3 | | | 16 | 0.2 | 0.3 |
| | | 32 | 0.0 | 0.0 | | | 32 | 0.0 | 0.0 |
| | 4 | 2 | 84.7 | 35.1 | | 4 | 2 | 86.2 | 49.2 |
| | | 4 | **86.5** | **49.2** | | | 4 | 82.5 | 59.3 |
| | | 8 | 72.1 | 39.2 | | | 8 | 64.7 | 42.5 |
| | | 16 | 25.2 | 13.1 | | | 16 | 13.5 | 10.0 |
| | | 32 | 0.2 | 0.4 | | | 32 | 0.1 | 0.2 |
| | 8 | 2 | 76.0 | 29.6 | | 8 | 2 | 84.7 | 44.3 |
| | | 4 | 82.8 | 39.9 | | | 4 | 85.3 | 59.2 |
| | | 8 | 81.4 | 43.9 | | | 8 | 80.7 | 57.1 |
| | | 16 | 66.5 | 32.7 | | | 16 | 60.3 | 41.0 |
| | | 32 | 20.2 | 9.9 | | | 32 | 9.7 | 7.5 |
| | 16 | 2 | 60.2 | 16.4 | | 16 | 2 | 77.7 | 40.6 |
| | | 4 | 77.1 | 30.6 | | | 4 | **86.6** | 54.6 |
| | | 8 | 83.5 | 40.5 | | | 8 | 86.2 | **60.3** |
| | | 16 | 82.4 | 45.3 | | | 16 | 81.0 | 59.0 |
| | | 32 | 70.2 | 35.8 | | | 32 | 58.7 | 42.1 |

## B.3 NEEDLE IN A HAYSTACK

Table 10 shows NIAH accuracy for Mamba2-2.7B Finetuned on either 4k or 8k sequence lengths using 25K examples. Results are shown across a grid of train/test segment combinations. Models trained with more segments generalize better to larger soup sizes during inference, with improvements especially pronounced at higher segment counts. In contrast, models trained on just 2 segments degrade sharply when evaluated on 8, 16, or 32 segments, highlighting the importance of training with sufficient segments for robust generalization in soup-based Finetuned models.

Table 11 reports NIAH accuracy for Mamba2-8B trained with only 7K examples at either 4k or 8k sequence length. Despite the reduced supervision and limited training (1 epoch), the 8B model achieves strong results and continues to exhibit the same trends as in higher-resource settings: soup-finetuned models outperform concat-finetuned baselines, and training with more segments improves generalization to higher test-time segment counts. In contrast, we found that the Mamba2-2.7B model did not converge reliably under this low-data setup. These results suggest that soupability and generalization benefits persist even in lower-resource regimes, but larger models are more robust to limited training data.

## B.4 RULER QA_1

We trained for 3 epochs and tested Mamba2-2.7B on the QA_1 subset of RULER, where each input includes 20 documents, only one of which contains the answer. This single-hop task tests the model's ability to isolate relevant information under extreme distraction. As shown in Table 12, `Soup w/ Average` trained on 2 segments slightly outperforms the concat baseline. While the margin is small and not statistically significant, this result demonstrates the potential of state souping even in sparse-relevance QA settings.

However, training on larger segment counts leads to worse performance, especially on smaller test-time soup sizes. Unlike multi-hop settings like HotpotQA, where broader exposure improves generalization, over-fragmentation in sparse single-hop tasks appears to dilute the signal.

Table 11: Evaluation accuracy (%) on NIAH task for 8B Mamba models Finetuned on 4k (left) or 8k (right) sequence length with 7K examples. Gray means Soup-finetuned results are better than respective Concat-finetuned results (72.55 / 14.2 for 4k, 66.55 / 18.7 for 8k). Bold represents the best in each column.

| **Finetuned on 4K sequence length** | | | | | **Finetuned on 8K sequence length** | | | | |
|---|---|---|---|---|---|---|---|---|---|
| **Method** | **Train Segments** | **Test Segments** | **Test Seq. Length** | | **Method** | **Train Segments** | **Test Segments** | **Test Seq. Length** | |
| | | | **4k** | **8k** | | | | **4k** | **8k** |
| Concat | 1 | 1 | 72.55 | 14.2 | Concat | 1 | 1 | 66.55 | 18.7 |
| Soup w/ Avg. | 2 | 2 | **80.4** | **30.2** | Soup w/ Avg. | 2 | 2 | 74.15 | **31.0** |
| | | 4 | 70.55 | 23.7 | | | 4 | 57.75 | 21.55 |
| | | 8 | 38.0 | 10.65 | | | 8 | 18.85 | 7.5 |
| | 4 | 2 | 79.4 | 27.7 | | 4 | 2 | **75.55** | **31.0** |
| | | 4 | 75.4 | 25.35 | | | 4 | 69.7 | 27.95 |
| | | 8 | 58.6 | 18.5 | | | 8 | 48.8 | 17.85 |
| | 8 | 2 | 72.2 | 23.2 | | 8 | 2 | 75.3 | 27.8 |
| | | 4 | 73.5 | 24.55 | | | 4 | 73.0 | 27.35 |
| | | 8 | 68.8 | 22.35 | | | 8 | 67.1 | 25.05 |

Table 12: EM / F1 scores on RULER QA_1 for Mamba2-2.7B trained and evaluated on 4k sequence length. Gray cells indicate performance exceeding the concat-finetuned test results (EM 48.4 / F1 64.8), and bold marks the highest test result of the task. Training on more segments improves generalization to higher evaluation segment counts until train with 20 segments. All experiments are run for 3 epochs due to limited data.

| **Method** | **Train Segments** | **Test Segments** | | | | |
|---|---|---|---|---|---|---|
| | | 1 | 2 | 5 | 10 | 20 |
| Concat | 1 | 48.4 / 64.8 | – | – | – | – |
| Soup w/ Average | 2 | – | **49.7 / 66.5** | 33.9 / 47.4 | 16.5 / 28.0 | 11.4 / 22.3 |
| | 5 | – | 46.2 / 60.4 | 44.0 / 59.1 | 26.5 / 39.3 | 17.3 / 29.4 |
| | 10 | – | 16.8 / 27.0 | 11.1 / 18.3 | 42.2 / 56.5 | 34.6 / 47.4 |
| | 20 | – | 19.4 / 30.4 | 15.2 / 25.4 | 24.9 / 37.2 | 24.6 / 38.4 |

# C  USE OF LARGE LANGUAGE MODELS

Large language models (LLMs) were utilized to assist in the preparation of this manuscript in two capacities. First, LLMs served as a writing aid for copy-editing and for improving the clarity and readability of the text. The authors drafted all original content and retained full editorial control, evaluating all suggestions to ensure they aligned with our intended meaning. Second, LLMs were used to support the literature discovery process. The authors provided seed papers on relevant topics and leveraged the models to find additional related work. All papers suggested by the LLMs were subsequently vetted by the authors for relevance before inclusion.

