# OpenReview forum: "The Surprising Soupability of Documents in State Space Models"
_ICLR.cc/2026/Conference — Submitted to ICLR 2026_

### Official Review · Reviewer_Ycg4 · 2025-10-19

**Soundness:** 2
**Presentation:** 1
**Contribution:** 1
**Rating:** 2
**Confidence:** 4

**Summary:**

The paper studies whether hidden states computed from multiple different documents can be composed post hoc, for downstream tasks such as QA and long-document reasoning, in SSMs (in particular Mamba-2). Towards this end, they propose the "souping" of document states, which are encoded independently, via commutative operators like averaging. The paper finds that a zero-shot approach is ineffective, and that "finetuning is crucial for unlocking soupability in pretrained SSMs".

**Strengths:**

The paper provides extensive experiments, across a variety of tasks, and scaling up to composing 256 documents.

The paper also studies how different fine-tuning approaches have varying degree of effects on the composability of the resulting states.

**Weaknesses:**

The paper's central investigation into 'document souping' is presented without acknowledging that this method was already introduced and evaluated as a baseline in [1], published in ICLR 2025. This omission is rather severe, as it overlooks the most direct prior work.

In particular, both papers are motivated by the exact same problem: the inefficiency of the monolithic, concatenation-based context processing in SSMs.  Their proposed solutions are conceptually almost identical -- (1) pre-processing individual documents by encoding them as fixed-size states, and (2) composing them in a commutative manner at inference time. Furthermore, the method proposed in [1] appears in fact superior, being theoretically-grounded and working better in both zero-shot and fine-tuned settings.

To elaborate, the core method of this work, which it terms "document souping" with simple averaging as the most effective operator, appears to be functionally identical to the "Soup" method that was not only studied in an even earlier paper [2] published on arXiv, but also explicitly implemented and evaluated as a **baseline** in [1]. What is even more concerning is that even though [2] is cited in this work, the fact that the proposed method here is identical appears heavily downplayed, with the only mention being:

> (Pióro et al., 2024) linearly combines task-specific hidden states for skill transfer. In contrast, our
method focuses on merging hidden states from disjoint document chunks, enabling compositional
reasoning across distributed corpora through simple aggregation strategies.

Consequently I strongly believe that this work not only fails to frame its contributions appropriately in the context of prior art, but also as a result is limited in its novelty and contributions beyond an empirical deep dive into an existing baseline.


[1] PICASO: Permutation-Invariant Context Composition with State Space Models. ICLR 2025.

[2] State Soup: In-Context Skill Learning, Retrieval and Mixing. arXiv 2024.

**Questions:**

See weaknesses

---

> ### Author Response · Authors · 2025-11-16
>
> We appreciate your time and feedback on our paper.
>
> Regarding [1], thank you for pointing this out, we indeed missed this paper despite what we believed was a very thorough search. However, we strongly feel that this work is orthogonal and complementary to ours. Specifically, [1] introduces a new, theoretically well-motivated method of aggregating SSM states that approximates (a permutation-invariant version of) full concatenation. They do not, however, conduct experiments on actual downstream tasks. No task accuracy or error metrics are reported; instead, they primarily present perplexity. The only accuracy scores we are aware of appear in Appendix B4, Table 2, and are used to show that fine-tuning does not deteriorate performance on general language understanding benchmarks, rather than studying the utility of document souping itself. This is fine for a paper that is mostly about the theoretical justification of a new method. However, it does not reveal the utility or specific limitations of document souping on real downstream tasks. One of our primary contributions is precisely that: we provide, to the best of our knowledge, the most comprehensive and thorough empirical investigation of document souping on actual tasks to date, including multi-hop reasoning.
>
> Regarding [2], as the reviewer points out, we were aware of this work and did cite it. However, we respectfully disagree that their work overlaps with ours. Neither we nor [2] claim to have invented “souping” i.e., the convex combination of vector representations (model parameters, model states, encodings, etc.). [2] demonstrates that in-context learning of skills can be souped. We instead demonstrate that factual knowledge in larger documents and corpora can be souped and even composed for multi-hop reasoning tasks. These contributions are complementary and distinct.
>
> We therefore believe that these existing works do not scoop or subsume what we have done. On the contrary, we view them very positively: now knowing about [1], we are eager to incorporate the theoretically justified aggregation method that they propose in our empirical evaluation.
>
> >"Soup" method that was not only studied in an even earlier paper [2] published on arXiv, but also explicitly implemented and evaluated as a baseline in [1].
>
> Finally, we would like to point out a subtle but important distinction between the actual methods we test and the baseline “souping” method in [1]. Our understanding is that in [1], inspired by [2], there is a baseline that attempts “souping” for a similar purpose. However, [1] appears to implement this only in a naive way without further details. In our experiments, we found that this naive approach does not work out of the box in zero-shot settings (where [1] primarily focuses on) for our downstream tasks. Our contribution is to make souping work effectively for them through an extensive set of experiments and ablations, such as fine-tuning Mamba2 models of different sizes (2.7B as well as 8B) under both decoder-only and encoder–decoder fine-tuning. Moreover, we experiment with operators other than averaging to study their effect as pooling strategies. In our paper, we show how souping performs for Multi-Doc QA, Long-Doc QA, and sparse-retrieval tasks, which, to our knowledge, has not been done before. We also demonstrate how this setup can extend to other architectures, such as Transformers (Section 3.3), which is referred to as future work in [1].
>
> Please let us know if you have any other concerns or questions, and we would be happy to discuss them.
>
> [1] PICASO: Permutation-Invariant Context Composition with State Space Models. ICLR 2025.
>
> [2] State Soup: In-Context Skill Learning, Retrieval and Mixing. arXiv 2024.

---

> ### Comment · Reviewer_Ycg4 · 2025-11-24
> **Framing of paper misrepresents its contribution**
>
> I thank the authors for their rebuttal and detailed response. However, after reviewing the response, I respectfully disagree with the author's feeling that [1] "is orthogonal and complementary to (this paper)", nor with authors' stance on [2] where they "respectfully disagree that (the) work overlaps with (theirs)".
>
> **The framing of the paper in its current form fundamentally misrepresents its contribution.**
>
> My key areas of concern in the paper's framing can be summarized by the title "The Surprising Soupability of Documents in State Space Models", and the contributions section "We introduce document souping, a mechanism for merging per-document hidden states" (L94).
>
> The *Soupability of Documents in State Space Models* has been demonstrated in [1]. The *Soupability* of hidden states in State Space Models have already been explored in [2]. Publishing a paper that presents existing works as a "surprising" novel contribution that "introduce(s) document souping" without explicitly stating it as such is very concerning.
>
> To elaborate on specific points from the rebuttal:
>
> > *[2] demonstrates that in-context learning of skills can be souped. We instead demonstrate that factual knowledge in larger documents and corpora can be souped and even composed for multi-hop reasoning tasks. These contributions are complementary and distinct.*
>
> Although the authors claim that this composition is done on "in-context learning of skills" rather than documents, the methodology is exactly the same albeit on a different dataset.
>
> More importantly, this exact mechanism applied to "documents and corpora" was already defined as a baseline in [1] ("Soup ... Simple averaging of states obtained from each context"), for per-document hidden states.
>
> > *[1] is "mostly about the theoretical justification of a new method" claiming it "does not reveal the utility... on real downstream tasks."*
> > *The only accuracy scores we are aware of appear in Appendix B4, Table 2,*
>
> This is factually incorrect based on the text of [1], which conducts experiments on OpenbookQA (multiple-choice QA) and MSMARCO (ranking / retrieval) - Tables 3 and 4.
>
> > *Our contribution is to make souping work effectively for them through an extensive set of experiments and ablations, such as fine-tuning Mamba2 models...*
>
> [1] explicit covers this in Sec. 5. The finding that "finetuning is crucial for unlocking soupability in pretrained SSMs" (L343) is also not a novel discovery of this paper; it is also a central result of [1].
>
> > *One of our primary contributions is precisely that: we provide, to the best of our knowledge, the most comprehensive and thorough empirical investigation of document souping on actual tasks to date, including multi-hop reasoning.*
>
> I acknowledge this point in the author's rebuttal, and this has also been listed as a strength of the paper in my original review.
>
> While I agree that empirical breadth is valuable, the paper in its current draft frames itself as the introduction of a novel method. To be considered for publication, the paper would need to be completely rewritten—including and especially the title—to frame it accurately as an extended empirical analysis of the "Soup" baseline from [1] applied to a comprehensive suite of evaluation tasks. Even with such a rewrite, the technical novelty would be limited. Therefore, I maintain my recommendation to reject.
>
> [1] PICASO: Permutation-Invariant Context Composition with State Space Models. ICLR 2025.
>
> [2] State Soup: In-Context Skill Learning, Retrieval and Mixing. arXiv 2024.

---

> > ### Author Response · Authors · 2025-12-02
> > **Framing of the paper and contributions**
> >
> > We appreciate the feedback and respond to the reviewer’s concerns below.
> >
> > As we mentioned in our initial response, we unfortunately missed [1]. However, after a thorough analysis and comparison of the work, we still believe that our work is complementary to [1] and [2], as it provides a detailed analysis of the souping mechanism with finetuning and contributes to a better community understanding of souping in State Space Models alongside [1] and [2]. Below, we outline the points that support this claim and respond to the reviewer's points:
> >
> > > The framing of the paper in its current form fundamentally misrepresents its contribution.
> >
> > After carefully rewording a few sentences in our manuscript and explicitly acknowledging [1], we have **uploaded a slightly revised version of the paper**. Only very minor changes were needed, as most of the paper focuses on how to effectively perform souping, achieve good performance with souping, study different options and tasks, and understand where this method might fail. As we mentioned, neither we nor [2] claim to have invented souping, as it is a well-known method for combining representations, hence the use of the term “souping” in all the aforementioned papers. Nevertheless, we have revised the few sentences in our manuscript that might have been misleading, explicitly acknowledged [1], which we had missed in our literature search, and added more information about [2]. We have also revised the title in the newly uploaded manuscript and will update it in the final version of the paper.
> >
> > > already defined as a baseline in [1]
> >
> > We acknowledge that our paper indeed focuses on the souping mechanism. However, we would like to respectfully emphasize that our contributions go beyond simply applying souping to documents and corpora, and instead focus on how to effectively perform souping, achieve good performance, study different options and tasks, and understand where this method might fail.
> >
> > **Neither [1] nor [2]** performs a detailed empirical analysis of document/corpus souping on realistic long-context benchmarks, its scaling behavior (e.g., number of documents, sequence lengths, and chunking strategies), or its behavior under alternative architectures such as Transformers.
> >
> > In [1], the "Soup" baseline is only one component within a broader method, and its empirical evaluation is very limited despite Tables 2 - 4 (which are themselves small and only appear in the appendix and no accuracy or task metric appears in the main paper), and it only reports experiments with the smaller 2.7B Mamba model. Moreover, souping of factual information for multi-hop composition, has not been demonstrated by [1] or, to the best of our knowledge, by any prior work: the small tables in the appendix of [1] consider a very simple setup, report only a couple of numbers, and are far from covering real-world QA reading comprehension. In contrast, our work is centered on systematically stress-testing this mechanism across multiple tasks and setups. Similarly, [2] focuses solely on in-context skill learning from souped states, rather than the document-level and multi-hop reasoning scenarios that we study.
> >
> > > [1] explicit covers this (finetuning) in Sec. 5.
> >
> > On finetuning, we agree that [1] also highlights the importance of finetuning for unlocking soupability in SSMs, but this is done specifically in the context of their PICASO method rather than for the "Soup" mechanism that we focus on. Our contribution here is to show, through a much more extensive set of experiments on document and corpus souping, how finetuning affects performance across tasks, architectures, and scaling regimes. In other words, our emphasis is not on the mere observation that finetuning is useful, but on characterizing when souping works, when it fails, and which design choices (e.g., finetuning strategy, pooling, number and length of documents, architecture) are critical in realistic, long-context settings.
> >
> > Overall, we see our work as providing, to the best of our knowledge, **the most comprehensive empirical study of document and corpus souping to date, spanning realistic benchmarks, a range of context lengths, and both SSMs and Transformers.** We believe that this systematic understanding of when souping succeeds or breaks down is valuable for the community, even if the high-level idea of souping representations has appeared before. Furthermore, together with the existing works [1] and [2], we believe that our paper will be a meaningful and complementary contribution for future work on souping in state space models.
> >
> > [1] PICASO: Permutation-Invariant Context Composition with State Space Models. ICLR 2025.
> >
> > [2] State Soup: In-Context Skill Learning, Retrieval and Mixing. arXiv 2024.

---

### Official Review · Reviewer_Yxuq · 2025-10-30

**Soundness:** 2
**Presentation:** 3
**Contribution:** 2
**Rating:** 4
**Confidence:** 4

**Summary:**

The paper addresses a problem of non-reusable representations of contextual corpora as a single fixed-length hidden state in the structured SSM. The proposed approach inspired by the model souping technique, allows the efficient procedure of encoding of contextual documents into reusable representations that can be pooled together for SSM reasoning. The experimental evaluations on multi-document and single long-document QA demonstrate that finetuned Mamba2 models with souped representations achieve competitive or superior performance compared to traditional concatenation-based encoding.

**Strengths:**

1. The paper is well-motivated and clearly presents the core challenge of single nonchangeable hidden state encoding for large contextual corpora for the task that supposes the modularity of the encoded representations.
2. The proposed mechanism is simple and well-described.
3. The superior computational efficiency of the proposed method over existing approaches is experimentally supported.

**Weaknesses:**

1. The limited generalizability and practical utility of the proposed method due to performance decrease in evaluations with the number of segments larger than used during training.
2. No statistical test or confidence intervals were reported to support the significance of the experimental results.
3. The paper lacks explicit comparisons of time and memory costs of the proposed method with document concatenation baselines to support the cost efficiency claim.
4. The paper does not provide any comparisons with related RAG methods.

**Questions:**

-

---

> ### Author Response · Authors · 2025-11-27
> **Part 1**
>
> Thank you for your careful review and constructive feedback. We are glad that you find our paper well-motivated and the experiments interesting. Below, we have given detailed answers to your questions.
>
> > The limited generalizability and practical utility of the proposed method due to performance decrease in evaluations with the number of segments larger than used during training.
>
> Generalization across different soup sizes is indeed a key consideration, and our results in Table 3 aim to address this. While performance tends to peak when the number of segments at test time matches the training configuration, models trained on k segments perform well when evaluated on fewer segments and generalize reasonably well to larger segments. For example, the model trained with 4 segments retains strong accuracy on 2 segments and maintains competitive performance even at 8 segments. This suggests that degradation from soup size mismatch is gradual rather than abrupt.
>
> Moreover, in Appendix B.1 SCALABILITY OF SOUP SIZES (and Table 8), we perform additional analysis on this matter, specifically, we show scalability results (EM %) on the NIAH dataset for a Mamba2-8B model with an 8k sequence length. We compare a model finetuned on 64 segments against a model that was subsequently trained further on 128 segments on up to 256 test segments. Continued training significantly improves performance on larger soup sizes, demonstrating the model’s capacity to adapt to wider contexts.
>
> Ideally, we believe that to achieve optimal performance across varying numbers of segments, the model could be trained using a mixture of segment configurations. This approach is conceptually aligned with how language models are often trained on a mixture of short and long sequences to support generalization across input lengths. A systematic exploration of this training strategy is a promising future direction to pursue.

---

> ### Author Response · Authors · 2025-11-27
> **Part 2**
>
> > No statistical test or confidence intervals were reported to support the significance of the experimental results.
>
> To address the reviewer’s concern, we have bootstrapped the complete results for RULER-NIAH (Table 3) and RULER-QA (Table 5) to show the significance of our souping-finetuned results compared to concat-finetuned results. Our procedure involved 5000 replications to estimate the 95% confidence interval; for each replication, we resampled the original test set data points with replacement. The ±k value denotes the maximum difference between the originally reported mean and either the upper or lower bound of the confidence interval. We will update all tables with confidence intervals in the final revised version.
> Here are the confidence-interval results based on Tables 3 and 5. Most intervals are within approximately ±2 points for Table 3 (finetuned on 4k/8k tokens and evaluated on 4k/8k tokens) and within approximately ±3 points for Table 5, for both exact match and F1 scores. These confidence intervals show that the best souping results are comparable to, and often outperform, the normal concat-finetuning results by a clear margin.
>
> **Table 3 Finetuned on 4K sequence length**
>
> | Method | Train Segments | Test Segments | Test Seq. Length (4k) | Test Seq. Length (8k) |
> | :--- | :---: | :---: | :---: | :---: |
> | **Concat** | 1 | 1 | 85.8 ± 1.6 | 24.25 ± 1.9 |
> | **Soup w/ Average** | 2 | 2 | 87.0 ± 1.5 | 38.45 ± 2.2 |
> | | 2 | 4 | 79.75 ± 1.8 | 32.05 ± 2.1 |
> | | 2 | 8 | 45.7 ± 2.2 | 16.3 ± 1.7 |
> | | 2 | 16 | 2.3 ± 0.7 | 0.8 ± 0.4 |
> | | 2 | 32 | 0.0 ± 0.0 | 0.0 ± 0.0 |
> | | 4 | 2 | 88.6 ± 1.4 | 40.7 ± 2.2 |
> | | 4 | 4 | 86.8 ± 1.5 | 38.7 ± 2.3 |
> | | 4 | 8 | 76.55 ± 1.9 | 29.3 ± 2.0 |
> | | 4 | 16 | 29.05 ± 2.1 | 11.45 ± 1.4 |
> | | 4 | 32 | 0.9 ± 0.4 | 0.5 ± 0.4 |
> | | 8 | 2 | 81.4 ± 1.8 | 34.25 ± 2.2 |
> | | 8 | 4 | 84.9 ± 1.6 | 36.85 ± 2.2 |
> | | 8 | 8 | 83.6 ± 1.7 | 36.45 ± 2.1 |
> | | 8 | 16 | 71.45 ± 2.0 | 28.9 ± 2.1 |
> | | 8 | 32 | 15.75 ± 1.7 | 9.25 ± 1.3 |
>
> **Table 3 Finetuned on 8K sequence length**
>
> | Method | Train Segments | Test Segments | Test Seq. Length (4k) | Test Seq. Length (8k) |
> | :--- | :---: | :---: | :---: | :---: |
> | **Concat** | 1 | 1 | 81.65 ± 1.8 | 35.55 ± 2.2 |
> | **Soup w/ Average** | 2 | 2 | **89.8** ± 1.3 | **48.4** ± 2.2 |
> | | 2 | 4 | 79.45 ± 1.8 | 38.15 ± 2.2 |
> | | 2 | 8 | 41.15 ± 2.3 | 15.45 ± 1.6 |
> | | 2 | 16 | 3.2 ± 0.8 | 1.45 ± 0.6 |
> | | 2 | 32 | 0.05 ± 0.1 | 0.0 ± 0.0 |
> | | 4 | 2 | 88.5 ± 1.4 | 46.55 ± 2.2 |
> | | 4 | 4 | 86.5 ± 1.5 | 43.85 ± 2.2 |
> | | 4 | 8 | 75.65 ± 1.9 | 34.6 ± 2.1 |
> | | 4 | 16 | 23.8 ± 1.9 | 11.5 ± 1.4 |
> | | 4 | 32 | 0.4 ± 0.3 | 0.55 ± 0.4 |
> | | 8 | 2 | 87.6 ± 1.5 | 45.3 ± 2.3 |
> | | 8 | 4 | 88.2 ± 1.5 | 45.85 ± 2.2 |
> | | 8 | 8 | 85.25 ± 1.6 | 43.9 ± 2.2 |
> | | 8 | 16 | 64.95 ± 2.1 | 32.25 ± 2.1 |
> | | 8 | 32 | 9.7 ± 1.3 | 7.9 ± 1.3 |
>
> **Table 5**
>
> | Method | Train Segments | 1 | 2 | 5 | 10 | 20 |
> | :--- | :---: | :--- | :--- | :--- | :--- | :--- |
> | **Concat** | 1 | 54.81 ± 3.7 / 71.90 ± 2.9 | -- | -- | -- | -- |
> | **Soup w/ Average** | 2 | -- | **58.38 ± 3.6 / 74.05 ± 2.8** | 34.89 ± 3.4 / 49.04 ± 3.2 | 13.19 ± 2.5 / 23.40 ± 2.7 | 12.36 ± 2.5 / 22.45 ± 2.7 |
> | | 5 | -- | **58.38 ± 3.6** / 73.89 ± 2.7 | 57.28 ± 3.6 / 72.09 ± 2.9 | 35.71 ± 3.4 / 50.80 ± 3.2 | 14.97 ± 2.8 / 26.25 ± 2.8 |
> | | 10 | -- | 21.02 ± 3.0 / 31.41 ± 3.0 | 13.87 ± 2.5 / 22.36 ± 2.7 | 52.75 ± 3.7 / 68.68 ± 2.9 | 43.82 ± 3.6 / 58.99 ± 3.2 |
> | | 20 | -- | 28.71 ± 3.3 / 41.44 ± 3.2 | 28.85 ± 3.3 / 42.93 ± 3.2 | 45.60 ± 3.7 / 61.78 ± 3.1 | 44.37 ± 3.7 / 60.85 ± 3.1 |
>
> > The paper lacks explicit comparisons of time and memory costs of the proposed method with document concatenation baselines to support the cost efficiency claim.
>
> Thanks for pointing this out.
>
> **Time.** As described in Section 4.4, Table 6 compares (i) the time required to process documents of length 2k–32k using full concatenation with (ii) the time required to load the souped hidden states and process only the input query. Our results show that our method is up to 14.6x faster than processing a fully concatenated 32k-token sequence.
>
> **Memory.** We appreciate the reviewer raising a potential point of confusion regarding memory usage at inference time. One key advantage of State Space Models (SSMs) is that their memory usage remains constant because the hidden state size is fixed, unlike Transformer-based models whose KV cache grows with context length. Although we did not explicitly emphasize this in Section 4.4, our method fully preserves this benefit as we only use a single souped hidden state. We thank the reviewer for pointing this out and will make the constant hidden-state-size property of SSMs explicit in this section and clarify it further in the revised version of the paper.

---

> ### Author Response · Authors · 2025-11-27
> **Part 3**
>
> > The paper does not provide any comparisons with related RAG methods.
>
> We would like to emphasize that document souping is not intended to replace RAG; rather, the two are complementary. One can first use RAG to retrieve a subset of relevant documents (e.g., book chapters) and then soup the hidden states corresponding to those retrieved documents. For example, in our HotPotQA experiments, we operate on two gold documents plus a set of distractors that are themselves obtained via a RAG-style retrieval step before being souped.
>
> We hope that our explanation have answered your questions and would be happy to discuss further.

---

### Official Review · Reviewer_g79h · 2025-10-31

**Soundness:** 3
**Presentation:** 3
**Contribution:** 4
**Rating:** 6
**Confidence:** 3

**Summary:**

This paper proposes to "soup" documents in SSM space and use them for various tasks. They show that although they cannot do them out of the box, SSMs can be trained to do so effectively. They show this in a variety of eval settings and with a variety of different ways/numbers of docs.

Overall, I think this paper is promising but I am concerned about the baselines. I am willing to improve my score if those are included (regardless of whether the results are better/worse than the proposed method).

**Strengths:**

- I think the topic of this paper is unique, both quite interesting and has surprisingly good results
- The proposed method seems to work across settings and as the number of documents increases
- The paper is well written and even includes an eval with standard transformers

**Weaknesses:**

1. My biggest concern is the lack of a FT'd baseline for these approaches. It seems a crucial comparison that is left out in favor for various different versions of the proposed technique (e.g. Table 1 and Table 3). It's also possible that I'm misunderstanding the baselines there, so please correct me if I'm wrong. Since the proposed version does FT'ing, I would expect all evals to have a baseline of the non-soup with fine-tuning.
2. [Minor] I think the experiment up to 256 docs needs to be in the main text, especially since it is referenced in the abstract. If accepted, I would strongly encourage the authors to do that.

**Questions:**

My questions are in the weaknesses.

---

> ### Author Response · Authors · 2025-11-16
>
> Thank you for taking the time to review our paper and for your positive remarks and valuable feedback! We are glad that you find the topic and our proposed document souping approach interesting and promising, and that the results across different settings come across clearly. Below, we provide detailed responses to your questions and concerns.
>
> > 1. My biggest concern is the lack of a FT'd baseline for these approaches. Since the proposed version does FT'ing, I would expect all evals to have a baseline of the non-soup with fine-tuning.
>
> We have, in fact, provided fine-tuned results for all approaches. In Table 1, we compare decoder-only finetuning and encoder–decoder finetuning, as well as different pooling strategies, to assess whether encoder–decoder finetuning is necessary or decoder-only finetuning suffices, and to select the most appropriate pooling strategy. In all other tables, our soup-finetuned results are reported alongside the concat-finetuned ones. For example, in Table 3, the heading “Finetuned on 4K sequence length” indicates that all results in the table are from finetuned models. The same holds for the remaining tables.
>
> > 2. [Minor] I think the experiment up to 256 docs needs to be in the main text, especially since it is referenced in the abstract. If accepted, I would strongly encourage the authors to do that.
>
> Thanks for bringing this to our attention. We agree that it would be useful to have these results in the main paper rather than the appendix. We will make this change in the final revision of our paper.
>
> We hope that our responses have addressed your concerns and provided additional clarity.

---

### Official Review · Reviewer_2Lgv · 2025-11-02

**Soundness:** 2
**Presentation:** 3
**Contribution:** 3
**Rating:** 4
**Confidence:** 3

**Summary:**

The paper introduces the method of document souping which combines hidden states from independently encoded documents in SSMs.  Each document is encoded using SSMs and the hidden representations are then pooled to be used a decoder conditioned on the pooled representations. The paper is then evaluated on Multi-Doc QA and Long Doc QA datasets, and outperforms the concat appraoch.

**Strengths:**

The paper shows the surprising result that the representations from SSMs can be pooled and used to answer question which allows for doucments to be parallely encoded offering speedups and cost savings. The paper shows the method scales to 256 documents and outperforms the concat appraoch on different datasets.  The idea and the results is interesting and would be of value.

**Weaknesses:**

One consideration that I would want the authors to test is instead of having random distractors, have 5-10 documents that each answer a different question and pool them together and test the singular representation on the different questions for the documents. My worry is that training on these documents (specifically the full finetuning) might have introduced some bias about what documents are more likely to answer questions or not and this will test if information from all of the documents is retained or not.

The other thing not explicitly tested/mentioned is 1) the scaling with length of the documents and 2) how long are the documents, is it equal to the sequence length of the model for each dataset i.e. can I soup together 16 4k token length documents?

**Questions:**

What is the average length of the encoded document?
What is the max length of the encoded documents is it equal to the max length of the model?
How well does the model preserve information across different documents souped together?

---

> ### Author Response · Authors · 2025-11-24
>
> Thank you for your time and valuable feedback! We are glad that you find our idea and results interesting and of value. We have tried to carefully answer your questions and concerns below:
>
> > One consideration to test is instead of having random distractors, have 5–10 documents that each answer a different question and pool them together and test the singular representation on the different questions for the documents. My worry is that training on these documents (specifically the full finetuning) might have introduced some bias
>
> If we understand correctly, your concern is whether the documents containing the answers have special features or patterns that distinguish them from the distractors, and whether, during finetuning, the model might learn to rely on these patterns and focus only on those documents. We believe that our experiments on datasets such as RACE address this concern. In RACE, a single passage is used to answer many questions, and different parts of the passage are relevant for different questions. We split these passages into chunks, compute their corresponding states, and then soup them. As shown in Table 4, the model is still able to extract the information needed to answer each question from the souped state. Please let us know if there are any further concerns about this potential bias or if we have misunderstood your question.
>
> We will also add an explanation about this in the paper as it is an interesting point and would be happy to run additional experiments if you think that would be useful.
>
> > (1) Scaling with length of the documents.
>
> We agree that it is important to measure how performance changes as document length / number of chunks increases while the chunk size is held fixed. We can already see some of this behavior in our existing results. Specifically, in Table 3 we have settings where, for example, we take a 4k-token document, split it into two 2k-token chunks, soup their states, and compare this to taking an 8k-token document, splitting it into four 2k-token chunks, and souping those. Similar comparisons hold for other configurations (e.g., splitting 4k into four chunks vs. 8k into eight chunks with the same per-chunk length). We believe these are the kinds of controlled comparisons (fixed chunk size, increasing number of chunks / effective document length) that the reviewer is asking about, and we show that the model remains robust as we scale up the total input length.
>
> We also want to note that the base Mamba2 2.7B [1] and 8B (non-hybrid) [2] models we use were both pretrained with a 4k context length. In this work we fine-tuned up to roughly twice this pretraining context size (8k), and scaling up further would be quite interesting but was unfortunately out of scope for our compute capabilities.
>
> Another place where we implicitly study scaling is in HotPotQA: adding distractors effectively increases the number of chunks and overall document length.
>
> We will make sure to include a more detailed explanation in the revised version of the paper to clarify this point.
>
> > (2) How long are the documents, is it equal to the sequence length of the model for each dataset? Can I soup together 16 4k token length documents? What is the average length of the encoded document? What is the max length of the encoded documents; is it equal to the max length of the model?
>
> In our different experiments, the length of the documents varies. In HotPotQA, each instance consists of multiple short passages (two gold documents plus a set of distractors), and we use this setting to study multi-hop reasoning and the effect of distractors for souping. In the other experiments, we increase the initial document length up to the base models’ context length (4k), chunk it into smaller segments, and then further experiment with sequences up to 2x the context length (8k).
>
> > How well does the model preserve information across different documents souped together?
>
> In our experiments, we have tested this in different setups, such as multi-hop reasoning (HotPotQA), long-document QA (RACE and related tasks), and NIAH-style settings, and we show that souping can achieve comparable performance to, and in many cases outperform, concatenation while also saving inference time. Moreover, in Table 8 of the appendix, we test souping up to 256 documents, to see how well it can preserve the information as the number of chunks increases.
>
> We hope that these explanations address your concerns. Please let us know if you have any additional questions; we would be happy to discuss them further or run additional experiments.
>
> [1] Transformers are SSMs: Generalized Models and Efficient Algorithms Through Structured State Space Duality. ICML 2024.
>
> [2] An Empirical Study of Mamba-based Language Models. arXiv 2024.

---

### Meta-Review · Area_Chair_2zXR · 2026-01-07

**Summary:**

While the author rebuttal has addressed many of the raised concerns, several issues remain that could substantially affect the final rating of the submission:

- A comparison with RAG-based methods is still necessary. The request for such experiments is not intended to assess whether the proposed approach can replace RAG, but rather to better situate the method within the existing landscape and clarify its relative strengths and value.

- Although the experiments on RACE partially address the concern, given the reviewers’ original intent, evaluating only on RACE and QuALITY does not constitute a standard experimental design in our community. Additional experiments involving true RAG setups on more recent long-context benchmarks would be valuable.

- The paper would benefit from a more thorough discussion of relevant prior work.

**Reviewer Concerns:**

I think Reviewer2's concerns are addressed.

**Reviewer Scores:**

The rest three reviewers may not change their scores.

---

### Decision · Program_Chairs · 2026-01-26

Reject